# *textTOvec*: Deep Contextualized Neural Autoregressive Topic Models of Language with Distributed Compositional Prior

**Pankaj Gupta[1,2], Yatin Chaudhary[1], Florian Buettner[1], Hinrich Schütze[2]**
[1]Corporate Technology, Machine-Intelligence (MIC-DE), Siemens AG Munich, Germany
[2]CIS, University of Munich (LMU) Munich, Germany
`{pankaj.gupta, yatin.chaudhary, buettner.florian}@siemens.com`

## Abstract

We address two challenges of probabilistic topic modelling in order to better estimate the probability of a word in a given context, i.e., $P(\text{word}|\text{context})$ : (1) *No language structure in context*: Probabilistic topic models ignore word order by summarizing a given context as a "bag-of-word" and consequently the semantics of words in the context is lost. In this work, we incorporate language structure by combining a neural autoregressive topic model (TM) (e.g., DocNADE) with a LSTM based language model (LSTM-LM) in a single probabilistic framework. The LSTM-LM learns a vector-space representation of each word by accounting for word order in local collocation patterns, while the TM simultaneously learns a latent representation from the entire document. In addition, the LSTM-LM models complex characteristics of language (e.g., syntax and semantics), while the TM discovers the underlying thematic structure in a collection of documents. We unite two complementary paradigms of learning the meaning of word occurrences by combining a topic model and a language model in a unified probabilistic framework, named as ctx-DocNADE. (2) *Limited context and/or smaller training corpus of documents*: In settings with a small number of word occurrences (i.e., lack of context) in short text or data sparsity in a corpus of few documents, the application of TMs is challenging. We address this challenge by incorporating external knowledge into neural autoregressive topic models via a language modelling approach: we use word embeddings as input of a LSTM-LM with the aim to improve the word-topic mapping on a smaller and/or short-text corpus. The proposed DocNADE extension is named as ctx-DocNADEe.

We present novel neural autoregressive topic model variants coupled with neural language models and embeddings priors that consistently outperform state-of-the-art generative topic models in terms of generalization (perplexity), interpretability (topic coherence) and applicability (retrieval and classification) over 7 long-text and 8 short-text datasets from diverse domains.

## 1 Introduction

Probabilistic topic models, such as LDA (Blei et al., 2003), Replicated Softmax (RSM) (Salakhutdinov & Hinton, 2009) and Document Neural Autoregressive Distribution Estimator (DocNADE) variants (Larochelle & Lauly, 2012; Zheng et al., 2016; Lauly et al., 2017; Gupta et al., 2019) are often used to extract topics from text collections, and predict the probabilities of each word in a given document belonging to each topic. Subsequently, they learn latent document representations that can be used to perform natural language processing (NLP) tasks such as information retrieval (IR), document classification or summarization. However, such probabilistic topic models ignore word order and represent a given context as a bag of its words, thereby disregarding semantic information.

To motivate our first task of extending probabilistic topic models to incorporate word order and language structure, assume that we conduct topic analysis on the following two sentences:

`Bear falls into market territory` and `Market falls into bear territory`

**Figure 1:** (left): A topic-word distribution due to global exposure, obtained from the matrix **W** as row-vector. (middle): Nearest neighbors in semantics space, represented by **W** in its column vectors. (right): BoW and cosine similarity illustration in distributed embedding space.

When estimating the probability of a word in a given context (here: $P(\text{"bear"}|\text{context})$), traditional topic models do not account for language structure since they ignore word order within the context and are based on "bag-of-words" (BoWs) only. In this particular setting, the two sentences have the same unigram statistics, but are about different topics. On deciding which topic generated the word "bear" in the second sentence, the preceding words "market falls" make it more likely that it was generated by a topic that assigns a high probability to words related to *stock market trading*, where "bear territory" is a colloquial expression in the domain. In addition, the language structure (e.g., syntax and semantics) is also ignored. For instance, the word "bear" in the first sentence is a proper noun and subject while it is an object in the second. In practice, topic models also ignore functional words such as "into", which may not be appropriate in some scenarios.

Recently, Peters et al. (2018) have shown that a deep contextualized LSTM-based language model (LSTM-LM) is able to capture different language concepts in a layer-wise fashion, e.g., the lowest layer captures language syntax and topmost layer captures semantics. However, in LSTM-LMs the probability of a word is a function of its sentence only and word occurrences are modeled in a *fine granularity*. Consequently, LSTM-LMs do not capture semantics at a document level. To this end, recent studies such as TDLM (Lau et al., 2017), Topic-RNN (Dieng et al., 2016) and TCNLM (Wang et al., 2018) have integrated the merits of latent topic and neural language models (LMs); however, they have focused on improving LMs with global (semantics) dependencies using latent topics.

Similarly, while bi-gram LDA based topic models (Wallach, 2006; Wang et al., 2007) and n-gram based topic learning (Lauly et al., 2017) can capture word order in short contexts, they are unable to capture long term dependencies and language concepts. In contrast, DocNADE variants (Larochelle & Lauly, 2012; Gupta et al., 2019) learns word occurrences across documents i.e., *coarse granularity* (in the sense that the topic assigned to a given word occurrence equally depends on all the other words appearing in the same document); however since it is based on the BoW assumption all language structure is ignored. In language modeling, Mikolov et al. (2010) have shown that recurrent neural networks result in a significant reduction of perplexity over standard n-gram models.

*Contribution 1*: We *introduce language structure* into neural autoregressive topic models via a LSTM-LM, thereby accounting for word ordering (or semantic regularities), language concepts and long-range dependencies. This allows for the accurate prediction of words, where the probability of each word is a function of global and local (semantics) contexts, modeled via DocNADE and LSTM-LM, respectively. The proposed neural topic model is named as *contextualized-Document Neural Autoregressive Distribution Estimator* (*ctx-DocNADE*) and offers learning complementary semantics by combining joint word and latent topic learning in a unified neural autoregressive framework. For instance, Figure 1 (left and middle) shows the complementary topic and word semantics, based on TM and LM representations of the term "fall". Observe that the topic captures the usage of "fall" in the context of *stock market trading*, attributed to the global (semantic) view.

While this is a powerful approach for incorporating language structure and word order in particular for long texts and corpora with many documents, learning from contextual information remains challenging in settings with short texts and few documents, since (1) limited word co-occurrences or little context (2) significant word non-overlap in such short texts and (3) small training corpus of documents lead to little evidence for learning word co-occurrences. However, distributional word representations (i.e. word embeddings) (Pennington et al., 2014) have shown to capture both the semantic and syntactic relatedness in words and demonstrated impressive performance in NLP tasks.

For example, assume that we conduct topic analysis over the two short text fragments: `Deal with stock index falls` and `Brace for market share drops`. Traditional topic models

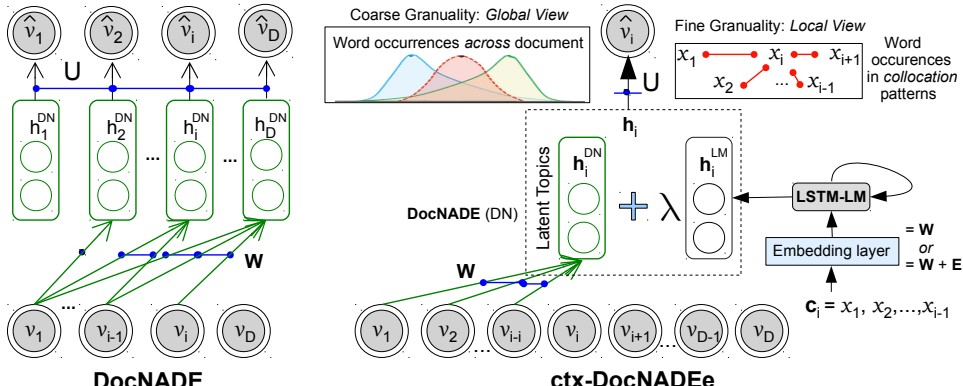

**Figure 2:** (left): DocNADE for the document $\mathbf{v}$. (right): ctx-DocNADEe for the observable corresponding to $v_i \in \mathbf{v}$. Blue colored lines signify the connections that share parameters. The observations (double circle) for each word $v_i$ are multinomial, where $v_i$ is the index in the vocabulary of the $i$th word of the document. $\mathbf{h}_i^{DN}$ and $\mathbf{h}_i^{LM}$ are hidden vectors from DocNADE and LSTM models, respectively for the target word $v_i$. Connections between each input $v_i$ and hidden units $\mathbf{h}_i^{DN}$ are shared. The symbol $\hat{v}_i$ represents the autoregressive conditionals $p(v_i|\mathbf{v}_{<i})$, computed using $\mathbf{h}_i$ which is a weighted sum of $\mathbf{h}_i^{DN}$ and $\mathbf{h}_i^{LM}$ in ctx-DocNADEe.

with "BoW" assumption will not be able to infer relatedness between word pairs such as (*falls*, *drops*) due to the lack of word-overlap and small context in the two phrases. However, in the distributed embedding space, the word pairs are semantically related as shown in Figure 1 (left).

Related work such as Sahami & Heilman (2006) employed web search results to improve the information in short texts and Petterson et al. (2010) introduced word similarity via thesauri and dictionaries into LDA. Das et al. (2015) and Nguyen et al. (2015) integrated word embeddings into LDA and Dirichlet Multinomial Mixture (DMM) (Nigam et al., 2000) models. Recently, Gupta et al. (2019) extends DocNADE by introducing pre-trained word embeddings in topic learning. However, they ignore the underlying language structure, e.g., word ordering, syntax, etc. In addition, DocNADE and its extensions outperform LDA and RSM topic models in terms of perplexity and IR.

*Contribution 2*: We incorporate *distributed compositional priors* in DocNADE: we use pre-trained word embeddings via LSTM-LM to supplement the multinomial topic model (i.e., DocNADE) in learning latent topic and textual representations on a smaller corpus and/or short texts. Knowing similarities in a distributed space and integrating this complementary information via a LSTM-LM, a topic representation is much more likely and coherent.

Taken together, we combine the advantages of complementary learning and external knowledge, and couple topic- and language models with pre-trained word embeddings to model short and long text documents in a unified neural autoregressive framework, named as *ctx-DocNADEe*. Our approach learns better textual representations, which we quantify via generalizability (e.g., perplexity), interpretability (e.g., topic extraction and coherence) and applicability (e.g., IR and classification).

To illustrate our two *contributions*, we apply our modeling approaches to 7 long-text and 8 short-text datasets from diverse domains and demonstrate that our approach consistently outperforms state-of-the-art generative topic models. Our learned representations, result in a gain of: (1) $4.6\%$ (.790 vs .755) in topic coherence, (2) $6.5\%$ (.615 vs .577) in precision at retrieval fraction 0.02, and (3) $4.4\%$ (.662 vs .634) in $F1$ for text classification, averaged over 6 long-text and 8 short-text datasets.

When applied to short-text and long-text documents, our proposed modeling approaches generate *contextualized topic vectors*, which we name *textTOvec*. The *code* is available at https://github.com/pgcool/textTOvec.

## 2 NEURAL AUTOREGRESSIVE TOPIC MODELS

Generative models are based on estimating the probability distribution of multidimensional data, implicitly requiring modeling complex dependencies. Restricted Boltzmann Machine (RBM) (Hinton et al., 2006) and its variants (Larochelle & Bengio, 2008) are probabilistic undirected models

of binary data. RSM (Salakhutdinov & Hinton, 2009) and its variants (Gupta et al., 2018) are generalization of the RBM, that are used to model word counts. However, estimating the complex probability distribution of the underlying high-dimensional observations is intractable. To address this challenge, NADE (Larochelle & Murray, 2011) decomposes the joint distribution of binary observations into autoregressive conditional distributions, each modeled using a feed-forward network. Unlike for RBM/RSM, this leads to tractable gradients of the data negative log-likelihood.

## 2.1 DOCUMENT NEURAL AUTOREGRESSIVE TOPIC MODEL (DOCNADE)

An extension of NADE and RSM, DocNADE (Larochelle & Lauly, 2012) models collections of documents as orderless bags of words (BoW approach), thereby disregarding any language structure. In other words, it is trained to learn word representations reflecting the underlying topics of the documents only, ignoring syntactical and semantic features as those encoded in word embeddings (Bengio et al., 2003; Mikolov et al., 2013; Pennington et al., 2014; Peters et al., 2018).

DocNADE (Lauly et al., 2017) represents a document by transforming its BoWs into a sequence $\mathbf{v} = [v_1, ..., v_D]$ of size $D$, where each element $v_i \in \{1, 2, ..., K\}$ corresponds to a multinomial observation (representing a word from a vocabulary of size $K$). Thus, $v_i$ is the index in the vocabulary of the $i$th word of the document $\mathbf{v}$. DocNADE models the joint distribution $p(\mathbf{v})$ of all words $v_i$ by decomposing it as $p(\mathbf{v}) = \prod_{i=1}^{D} p(v_i|\mathbf{v}_{<i})$, where each autoregressive conditional $p(v_i|\mathbf{v}_{<i})$ for the word observation $v_i$ is computed using the preceding observations $\mathbf{v}_{<i} \in \{v_1, ..., v_{i-1}\}$ in a feed-forward neural network for $i \in \{1, ...D\}$,

$$\mathbf{h}_i^{DN}(\mathbf{v}_{<i}) = g(\mathbf{e} + \sum_{k<i} \mathbf{W}_{:,v_k}) \text{ and } p(v_i = w|\mathbf{v}_{<i}) = \frac{\exp(b_w + \mathbf{U}_{w,:}\mathbf{h}_i^{DN}(\mathbf{v}_{<i}))}{\sum_{w'} \exp(b_{w'} + \mathbf{U}_{w',:}\mathbf{h}_i^{DN}(\mathbf{v}_{<i}))} \tag{1}$$

where $g(\cdot)$ is an activation function, $\mathbf{U} \in \mathbb{R}^{K \times H}$ is a weight matrix connecting hidden to output, $\mathbf{e} \in \mathbb{R}^H$ and $\mathbf{b} \in \mathbb{R}^K$ are bias vectors, $\mathbf{W} \in \mathbb{R}^{H \times K}$ is a word representation matrix in which a column $\mathbf{W}_{:,v_i}$ is a vector representation of the word $v_i$ in the vocabulary, and $H$ is the number of hidden units (topics). The log-likelihood of any document $\mathbf{v}$ of any arbitrary length is given by: $\mathcal{L}^{DN}(\mathbf{v}) = \sum_{i=1}^{D} \log p(v_i|\mathbf{v}_{<i})$. Note that the past word observations $\mathbf{v}_{<i}$ are orderless due to BoWs, and may not correspond to the words preceding the $i$th word in the document itself.

---

**Algorithm 1** Computation of $\log p(\mathbf{v})$

    **Input**: A training document $\mathbf{v}$
    **Input**: Word embedding matrix $\mathbf{E}$
    **Output**: $\log p(\mathbf{v})$
1: $\mathbf{a} \leftarrow \mathbf{e}$
2: $q(\mathbf{v}) = 1$
3: **for** $i$ from 1 to $D$ **do**
4:     compute $\mathbf{h}_i$ and $p(v_i|\mathbf{v}_{<i})$
5:     $q(\mathbf{v}) \leftarrow q(\mathbf{v})p(v_i|\mathbf{v}_{<i})$
6:     $\mathbf{a} \leftarrow \mathbf{a} + \mathbf{W}_{:,v_i}$
7: $\log p(\mathbf{v}) \leftarrow \log q(\mathbf{v})$

| model | $\mathbf{h}_i$ | $p(v_i\|\mathbf{v}_{<i})$ |
|---|---|---|
| DocNADE | $\mathbf{h}_i^{DN} \leftarrow g(\mathbf{a})$ 
 $\mathbf{h}_i \leftarrow \mathbf{h}_i^{DN}$ | equation 1 |
| ctx-DocNADE | $\mathbf{h}_i^{LM} \leftarrow \text{LSTM}(\mathbf{c}_i, \text{embedding} = \mathbf{W})$ 
 $\mathbf{h}_i \leftarrow \mathbf{h}_i^{DN} + \lambda\,\mathbf{h}_i^{LM}$ | equation 2 |
| ctx-DocNADEe | $\mathbf{h}_i^{LM} \leftarrow \text{LSTM}(\mathbf{c}_i, \text{embedding} = \mathbf{W} + \mathbf{E})$ 
 $\mathbf{h}_i \leftarrow \mathbf{h}_i^{DN} + \lambda\,\mathbf{h}_i^{LM}$ | equation 2 |

**Table 1:** Computation of $\mathbf{h}_i$ and $p(v_i|\mathbf{v}_{<i})$ in DocNADE, ctx-DocNADE and ctx-DocNADEe models, correspondingly used in estimating $\log p(\mathbf{v})$ (Algorithm 1).

## 2.2 DEEP CONTEXTUALIZED DOCNADE WITH DISTRIBUTIONAL SEMANTICS

We propose two extensions of the DocNADE model: (1) *ctx-DocNADE*: introducing language structure via LSTM-LM and (2) *ctx-DocNADEe*: incorporating external knowledge via pre-trained word embeddings $\mathbf{E}$, to model short and long texts. The unified network(s) account for the ordering of words, syntactical and semantic structures in a language, long and short term dependencies, as well as external knowledge, thereby circumventing the major drawbacks of BoW-based representations.

Similar to DocNADE, ctx-DocNADE models each document $\mathbf{v}$ as a sequence of multinomial observations. Let $[x_1, x_2, ..., x_N]$ be a sequence of N words in a given document, where $x_i$ is represented by an embedding vector of dimension, *dim*. Further, for each element $v_i \in \mathbf{v}$, let $\mathbf{c}_i = [x_1, x_2, ..., x_{i-1}]$ be the context (preceding words) of $i$th word in the document. Unlike in DocNADE, the conditional probability of the word $v_i$ in ctx-DocNADE (or ctx-DocNADEe) is a function of two hidden vectors: $\mathbf{h}_i^{DN}(\mathbf{v}_{<i})$ and $\mathbf{h}_i^{LM}(\mathbf{c}_i)$, stemming from the DocNADE-based and

| | short-text | | | | | | | | | long-text | | | | | | | |
|---|---|---|---|---|---|---|---|---|---|---|---|---|---|---|---|---|---|
| **Data** | **Train** | **Val** | **Test** | **\|RV\|** | **\|FV\|** | **L** | **C** | **Domain** | **Data** | **Train** | **Val** | **Test** | **\|RV\|** | **\|FV\|** | **L** | **C** | **Domain** |
| 20NSshort | 1.3k | 0.1k | 0.5k | 1.4k | 1.4k | 13.5 | 20 | News | 20NSsmall | 0.4k | 0.2k | 0.2k | 2k | 4555 | 187.5 | 20 | News |
| TREC6 | 5.5k | 0.5k | 0.5k | 2k | 2295 | 9.8 | 6 | Q&A | Reuters8 | 5.0k | 0.5k | 2.2k | 2k | 7654 | 102 | 8 | News |
| R21578title† | 7.3k | 0.5k | 3.0k | 2k | 2721 | 7.3 | 90 | News | 20NS | 7.9k | 1.6k | 5.2k | 2k | 33770 | 107.5 | 20 | News |
| Subjectivity | 8.0k | .05k | 2.0k | 2k | 7965 | 23.1 | 2 | Senti | R21578† | 7.3k | 0.5k | 3.0k | 2k | 11396 | 128 | 90 | News |
| Polarity | 8.5k | .05k | 2.1k | 2k | 7157 | 21.0 | 2 | Senti | BNC | 15.0k | 1.0k | 1.0k | 9.7k | 41370 | 1189 | - | News |
| TMNtitle | 22.8k | 2.0k | 7.8k | 2k | 6240 | 4.9 | 7 | News | SiROBs† | 27.0k | 1.0k | 10.5k | 3k | 9113 | 39 | 22 | Indus |
| TMN | 22.8k | 2.0k | 7.8k | 2k | 12867 | 19 | 7 | News | AGNews | 118k | 2.0k | 7.6k | 5k | 34071 | 38 | 4 | News |
| AGnewstitle | 118k | 2.0k | 7.6k | 5k | 17125 | 6.8 | 4 | News | | | | | | | | | |

**Table 2:** Data statistics: Short/long texts and/or small/large corpora from diverse domains. Symbols- Avg: average, $L$: avg text length (#words), $|RV|$ and $|FV|$: size of reduced (RV) and full vocabulary (FV), $C$: number of classes, Senti: Sentiment, Indus: Industrial, 'k':thousand and †: multi-label. For short-text, $L<25$.

LSTM-based components of ctx-DocNADE, respectively:

$$\mathbf{h}_i(\mathbf{v}_{<i}) = \mathbf{h}_i^{DN}(\mathbf{v}_{<i}) + \lambda \ \mathbf{h}_i^{LM}(\mathbf{c}_i) \text{ and } p(v_i = w|\mathbf{v}_{<i}) = \frac{\exp(b_w + \mathbf{U}_{w,:}\mathbf{h}_i(\mathbf{v}_{<i}))}{\sum_{w'} \exp(b_{w'} + \mathbf{U}_{w',:}\mathbf{h}_i(\mathbf{v}_{<i}))} \tag{2}$$

where $\mathbf{h}_i^{DN}(\mathbf{v}_{<i})$ is computed as in DocNADE (equation 1) and $\lambda$ is the mixture weight of the LM component, which can be optimized during training (e.g., based on the validation set). The second term $\mathbf{h}_i^{LM}$ is a context-dependent representation and output of an LSTM layer at position $i-1$ over input sequence $\mathbf{c}_i$, trained to predict the next word $v_i$. The LSTM offers history for the $i$th word via modeling temporal dependencies in the input sequence, $\mathbf{c}_i$. The conditional distribution for each word $v_i$ is estimated by equation 2, where the unified network of DocNADE and LM combines global and context-dependent representations. Our model is jointly optimized to maximize the pseudo log likelihood, $\log p(\mathbf{v}) \approx \sum_{i=1}^{D} \log p(v_i|\mathbf{v}_{<i})$ with stochastic gradient descent. See Larochelle & Lauly (2012) for more details on training from bag of word counts.

In the weight matrix $\mathbf{W}$ of DocNADE (Larochelle & Lauly, 2012), each row vector $\mathbf{W}_{j,:}$ encodes topic information for the $j$th hidden topic feature and each column vector $\mathbf{W}_{:,v_i}$ is a vector for the word $v_i$. To obtain complementary semantics, we exploit this property and expose $\mathbf{W}$ to both global and local influences by sharing $\mathbf{W}$ in the DocNADE and LSTM-LM componenents. Thus, the embedding layer of LSTM-LM component represents the column vectors.

*ctx-DocNADE*, in this realization of the unified network the embedding layer in the LSTM component is randomly initialized. This extends DocNADE by accounting for the ordering of words and language concepts via context-dependent representations for each word in the document.

*ctx-DocNADEe*, the second version extends ctx-DocNADE with distributional priors, where the embedding layer in the LSTM component is initialized by the sum of a pre-trained embedding matrix $\mathbf{E}$ and the weight matrix $\mathbf{W}$. Note that $\mathbf{W}$ is a model parameter; however $\mathbf{E}$ is a static prior.

Algorithm 1 and Table 1 show the $\log p(\mathbf{v})$ for a document $\mathbf{v}$ in three different settings: *DocNADE*, *ctx-DocNADE* and *ctx-DocNADEe*. In the DocNADE component, since the weights in the matrix $\mathbf{W}$ are tied, the linear activation $\mathbf{a}$ can be re-used in every hidden layer and computational complexity reduces to $O(HD)$, where H is the size of each hidden layer. In every epoch, we run an LSTM over the sequence of words in the document and extract hidden vectors $h_i^{LM}$, corresponding to $\mathbf{c}_i$ for every target word $v_i$. Therefore, the computational complexity in ctx-DocNADE or ctx-DocNADEe is $O(HD + \mathfrak{N})$, where $\mathfrak{N}$ is the total number of edges in the LSTM network (Hochreiter & Schmidhuber, 1997; Sak et al., 2014). The trained models can be used to extract a *textTOvec* representation, i.e., $\mathbf{h}(\mathbf{v}^*) = \mathbf{h}^{DN}(\mathbf{v}^*) + \lambda \mathbf{h}^{LM}(\mathbf{c}_{N+1}^*)$ for the text $\mathbf{v}^*$ of length $\mathbf{D}^*$, where $\mathbf{h}^{DN}(\mathbf{v}^*) = g(\mathbf{e} + \sum_{k \leq \mathbf{D}^*} \mathbf{W}_{:,v_k})$ and $\mathbf{h}^{LM}(\mathbf{c}_{N+1}^*) = \text{LSTM}(\mathbf{c}_{N+1}^*, \text{embedding} = \mathbf{W} \text{ or } (\mathbf{W} + \mathbf{E}))$.

*ctx-DeepDNEe*: DocNADE and LSTM can be extended to a deep, multiple hidden layer architecture by adding new hidden layers as in a regular deep feed-forward neural network, allowing for improved performance. In the deep version, the first hidden layer is computed in an analogous fashion to DocNADE variants (equation 1 or 2). Subsequent hidden layers are computed as:

$$\mathbf{h}_{i,d}^{DN}(\mathbf{v}_{<i}) = g(\mathbf{e}_d + \mathbf{W}_{i,d} \cdot \mathbf{h}_{i,d-1}(\mathbf{v}_{<i})) \ \text{ or } \ \mathbf{h}_{i,d}^{LM}(\mathbf{c}_i) = deepLSTM(\mathbf{c}_i, \text{depth} = d)$$

for $d = 2, ...n$, where n is the total number of hidden layers (i.e., depth) in the deep feed-forward and LSTM networks. For $d$=1, the hidden vectors $\mathbf{h}_{i,1}^{DN}$ and $\mathbf{h}_{i,1}^{LM}$ correspond to equations 1 and 2. The conditional $p(v_i = w|\mathbf{v}_{<i})$ is computed using the last layer $n$, i.e., $\mathbf{h}_{i,n} = \mathbf{h}_{i,n}^{DN} + \lambda \mathbf{h}_{i,n}^{LM}$.

| Model | 20NSshort | | TREC6 | | R21578title | | Subjectivity | | Polarity | | TMNtitle | | TMN | | AGnewstitle | | Avg | |
|---|---|---|---|---|---|---|---|---|---|---|---|---|---|---|---|---|---|---|
| | IR | $F1$ | IR | $F1$ | IR | $F1$ | IR | $F1$ | IR | $F1$ | IR | $F1$ | IR | $F1$ | IR | $F1$ | IR | $F1$ |
| *glove*(RV) | .236 | .493 | .480 | .798 | .587 | .356 | .754 | .882 | .543 | .715 | .513 | .693 | .638 | .736 | .588 | 814 | .542 | .685 |
| *glove*(FV) | .236 | .488 | .480 | .785 | .595 | .356 | .775 | .901 | .553 | .728 | .545 | .736 | .643 | .813 | .612 | .830 | .554 | .704 |
| *doc2vec* | .090 | .413 | .260 | .400 | .518 | .176 | .571 | .763 | .510 | .624 | .190 | .582 | .220 | .720 | .265 | .600 | .328 | .534 |
| *Gauss-LDA* | .080 | .118 | .325 | .202 | .367 | .012 | .558 | .676 | .505 | .511 | .408 | .472 | .713 | .692 | .516 | .752 | .434 | .429 |
| *glove-DMM* | .183 | .213 | .370 | .454 | .273 | .011 | .738 | .834 | .515 | .585 | .445 | .590 | .551 | .666 | .540 | .652 | .451 | .500 |
| *glove-LDA* | .160 | .320 | .300 | .600 | .387 | .052 | .610 | .805 | .517 | .607 | .260 | .412 | .428 | .627 | .547 | .687 | .401 | .513 |
| *TDLM* | .219 | .308 | .521 | .671 | .563 | .174 | .839 | .885 | .520 | .599 | .535 | .657 | .672 | .767 | .534 | .722 | .550 | .586 |
| *DocNADE(RV)* | .290 | .440 | .550 | .804 | .657 | .313 | .820 | .889 | .560 | .699 | .524 | .664 | .652 | .759 | .656 | .819 | .588 | .673 |
| *DocNADE(FV)* | .290 | .440 | .546 | .791 | .654 | .302 | .848 | .907 | .576 | .724 | .525 | .688 | .687 | .796 | .678 | .821 | .600 | .683 |
| *DeepDNE* | .100 | .080 | .479 | .629 | .630 | .221 | .865 | .909 | .517 | .531 | .536 | .661 | .671 | .783 | .682 | .825 | .558 | .560 |
| *ctx-DocNADE* | .296 | .440 | .595 | .817 | .641 | .300 | .874 | .910 | .591 | .725 | .560 | .687 | .692 | .793 | .691 | .826 | .617 | .688 |
| *ctx-DocNADEe* | **.306** | .490 | .599 | .824 | **.656** | .308 | .874 | .917 | **.605** | .740 | **.595** | .726 | **.698** | .806 | **.703** | .828 | **.630** | .705 |
| *ctx-DeepDNEe* | .278 | .416 | **.606** | .804 | .647 | .244 | **.878** | .920 | .591 | .723 | .576 | .694 | .687 | .796 | .689 | .826 | .620 | .688 |

**Table 3:** State-of-the-art comparison: IR (i.e, IR-precision at 0.02 fraction) and classification $F1$ for *short* texts, where $Avg$: average over the row values, the **bold** and underline: the maximum for IR and F1, respectively.

# 3 EVALUATION

We apply our modeling approaches (in improving topic models, i.e, DocNADE using language concepts from LSTM-LM) to 8 short-text and 7 long-text datasets of varying size with single/multi-class labeled documents from public as well as industrial corpora. We present four quantitative measures in evaluating topic models: generalization (perplexity), topic coherence, text retrieval and categorization. See the *appendices* for the data description and example texts. Table 2 shows the data statistics, where 20NS and R21578 signify 20NewsGroups and Reuters21578, respectively.

*Baselines*: While, we evaluate our multi-fold contributions on four tasks: generalization (perplexity), topic coherence, text retrieval and categorization, we compare performance of our proposed models `ctx-DocNADE` and `ctx-DocNADEe` with related baselines based on: **(1)** word representation: `glove` (Pennington et al., 2014), where a document is represented by summing the embedding vectors of it's words, **(2)** document representation: `doc2vec` (Le & Mikolov, 2014), **(3)** LDA based BoW TMs: `ProdLDA` (Srivastava & Sutton, 2017) and `SCHOLAR`[1] (Card et al., 2017) **(4)** neural BoW TMs: `DocNADE` and `NTM` (Cao et al., 2015) and , **(5)** TMs, including pre-trained word embeddings: `Gauss-LDA` (`GaussianLDA`) (Das et al., 2015), and `glove-DMM`, `glove-LDA` (Nguyen et al., 2015). **(6)** jointly[2] trained topic and language models: `TDLM` (Lau et al., 2017), `Topic-RNN` (Dieng et al., 2016) and `TCNLM` (Wang et al., 2018).

| Model | 20NSsmall | | Reuters8 | | 20NS | | R21578 | | SiROBs | | AGnews | | Avg | |
|---|---|---|---|---|---|---|---|---|---|---|---|---|---|---|
| | IR | $F1$ | IR | $F1$ | IR | $F1$ | IR | $F1$ | IR | $F1$ | IR | $F1$ | IR | $F1$ |
| *glove*(RV) | .214 | .442 | .845 | .830 | .200 | .608 | .644 | .316 | .273 | .202 | .725 | .870 | .483 | .544 |
| *glove*(FV) | .238 | .494 | .837 | .880 | .253 | .632 | .659 | .340 | .285 | .217 | .737 | .890 | .501 | .575 |
| *doc2vec* | .200 | .450 | .586 | .852 | .216 | .691 | .524 | .215 | .282 | .226 | .387 | .713 | .365 | .524 |
| *Gauss-LDA* | .090 | .080 | .712 | .557 | .142 | .340 | .539 | .114 | .232 | .070 | .456 | .818 | .361 | .329 |
| *glove-DMM* | .060 | .134 | .623 | .453 | .092 | .187 | .501 | .023 | .226 | .050 | - | - | - | - |
| *DocNADE(RV)* | .270 | .530 | **.884** | .890 | .366 | .644 | **.723** | .336 | .374 | .298 | .787 | .882 | .567 | .596 |
| *DocNADE(FV)* | .299 | .509 | .879 | .907 | .427 | .727 | .715 | .340 | .382 | .308 | .794 | .888 | .582 | .613 |
| *ctx-DocNADE* | .313 | .526 | .880 | .898 | .472 | .732 | .714 | .315 | .386 | .309 | .791 | .890 | .592 | .611 |
| *ctx-DocNADEe* | **.327** | .524 | .883 | .900 | **.486** | .745 | .721 | .332 | **.390** | .311 | **.796** | .894 | **.601** | .618 |

**Table 4:** IR-precision at fraction 0.02 and classification $F1$ for *long* texts

| Model | PPL | | Model | PPL |
|---|---|---|---|---|
| DocNADE | 980 | *Subjectivity* | DocNADE | 846 |
| ctx-DocNADE | 968 | *AGnewstitle* | ctx-DocNADE | 822 |
| ctx-DocNADEe | 966 | | ctx-DocNADEe | 820 |
| DocNADE | 283 | *Reuters8* | DocNADE | 1375 |
| ctx-DocNADE | 276 | *20NS* | ctx-DocNADE | 1358 |
| ctx-DocNADEe | 272 | | ctx-DocNADEe | 1361 |
| DocNADE | 1437 | *TMNtitle* | DocNADE | 646 |
| ctx-DocNADE | 1430 | *20NSshort* | ctx-DocNADE | 656 |
| ctx-DocNADEe | 1427 | | ctx-DocNADEe | 648 |

**Table 5:** Generalization: PPL

*Experimental Setup*: DocNADE is often trained on a reduced vocabulary (RV) after pre-processing (e.g., ignoring functional words, etc.); however, we also investigate training it on full text/vocabulary (FV) (Table 2) and compute document representations to perform different evaluation tasks. The FV setting preserves the language structure, required by LSTM-LM, and allows a fair comparison of DocNADE+FV and ctx-DocNADE variants. We use the glove embedding of 200 dimensions. All the baselines and proposed models (*ctx-DocNADE*, *ctx-DocNADEe* and *ctx-DeepDNEe*) were run in the FV setting over 200 topics to quantify the quality of the learned representations. To better initialize the complementary learning in ctx-DocNADEs, we perform a pre-training for 10 epochs with $\lambda$ set to 0. See the *appendices* for the experimental setup and hyperparameters for the following tasks, including the ablation over $\lambda$ on validation set.

---

[1]focuses on incorporating meta-data (author, date, etc.) into TMs; SCHOLAR w/o meta-data $\equiv$ ProdLDA
[2]though focused on improving language models using topic models, different to our motivation

| Data | glove-DMM | | glove-LDA | | DocNADE | | ctx-DNE | | ctx-DNEe | | Data | glove-DMM | | DocNADE | | ctx-DNE | | ctx-DNEe | |
|---|---|---|---|---|---|---|---|---|---|---|---|---|---|---|---|---|---|---|---|
| | W10 | W20 | W10 | W20 | W10 | W20 | W10 | W20 | W10 | W20 | | W10 | W20 | W10 | W20 | W10 | W20 | W10 | W20 |
| *20NSshort* | .512 | .575 | .616 | .767 | .669 | .779 | .682 | .794 | **.696** | **.801** | *Subjectivity* | .538 | .433 | .613 | .749 | .629 | .767 | **.634** | **.771** |
| *TREC6* | .410 | .475 | .551 | .736 | .699 | .818 | **.714** | **.810** | .713 | .809 | *AGnewstitle* | .584 | .678 | .731 | .757 | .739 | .858 | **.746** | **.865** |
| *R21578title* | .364 | .458 | .478 | .677 | .701 | .812 | .713 | .802 | **.723** | **.834** | *20NSsmall* | **.578** | .548 | .508 | .628 | .546 | .667 | .565 | **.692** |
| *Polarity* | .637 | .363 | .375 | .468 | .610 | .742 | .611 | .756 | **.650** | **.779** | *Reuters8* | .372 | .302 | .583 | .710 | .584 | .710 | **.592** | **.714** |
| *TMNtitle* | .633 | .778 | .651 | .798 | .712 | .822 | .716 | .831 | **.735** | **.845** | *20NS* | .458 | .374 | .606 | .729 | .615 | .746 | **.631** | **.759** |
| *TMN* | .705 | .444 | .550 | .683 | .642 | .762 | .639 | .759 | **.709** | **.825** | *Avg (all)* | .527 | .452 | .643 | .755 | .654 | .772 | **.672** | **.790** |

**Table 6:** Average coherence for *short* and *long* texts over 200 topics in FV setting, where *DocNADE ↔ DNE*

We run TDLM[3] (Lau et al., 2017) for all the short-text datasets to evaluate the quality of representations learned in the spare data setting. For a fair comparison, we set 200 topics and hidden size, and initialize with the same pre-trained word embeddings (i.e., glove) as used in the ctx-DocNADEe.

## 3.1 GENERALIZATION: PERPLEXITY (PPL)

To evaluate the generative performance of the topic models, we estimate the log-probabilities for the test documents and compute the average held-out perplexity ($PPL$) per word as, $PPL = \exp\left(-\frac{1}{z}\sum_{t=1}^{z}\frac{1}{|\mathbf{v}^t|}\log p(\mathbf{v}^t)\right)$, where $z$ and $|\mathbf{v}^t|$ are the total number of documents and words in a document $\mathbf{v}^t$. For DocNADE, the log-probability $\log p(\mathbf{v}^t)$ is computed using $\mathcal{L}^{DN}(\mathbf{v})$; however, we ignore the mixture coefficient, i.e., $\lambda=0$ (equation 2) to compute the exact log-likelihood in ctx-DocNADE versions. The optimal $\lambda$ is determined based on the validation set. Table 5 quantitatively shows the PPL scores, where the complementary learning with $\lambda = 0.01$ (optimal) in ctx-DocNADE achieves lower perplexity than the baseline DocNADE for both short and long texts, e.g., (822 vs 846) and (1358 vs 1375) on *AGnewstitle* and *20NS*[4] datasets, respectively in the FV setting.

## 3.2 INTERPRETABILITY: TOPIC COHERENCE

We compute topic coherence (Chang et al., 2009; Newman et al., 2009; Gupta et al., 2018) to assess the meaningfulness of the underlying topics captured. We choose the coherence measure proposed by Röder et al. (2015) , which identifies context features for each topic word using a sliding window over the reference corpus. Higher scores imply more coherent topics.

We use the gensim module (*radimrehurek.com/gensim/models/coherencemodel.html*, *coherence type* = *c_v*) to estimate coherence for each of the 200 topics (top 10 and 20 words). Table 6 shows average coherence over 200 topics, where the higher scores in ctx-DocNADE compared to DocNADE (.772 vs .755) suggest that the contextual information and language structure help in generating more coherent topics. The introduction of embeddings in ctx-DocNADEe boosts the topic coherence, leading to a gain of 4.6% (.790 vs .755) on average over 11 datasets. Note that the proposed models also outperform the baselines methods glove-DMM and glove-LDA. Qualitatively, Table 8 illustrates an example topic from the 20NSshort text dataset for DocNADE, ctx-DocNADE and ctx-DocNADEe, where the inclusion of embeddings results in a more coherent topic.

*Additional Baslines*: We further compare our proposed models to other approaches that combining topic and language models, such as TDLM (Lau et al., 2017), Topic-RNN (Dieng et al., 2016) and TCNLM (Wang et al., 2018). However, the related studies focus on improving language models using topic models: in contrast, the focus of our work is on improving topic models for textual representations (short-text or long-text documents) by incorporating language concepts (e.g., word ordering, syntax, semantics, etc.) and external knowledge (e.g., word embeddings) via neural language models, as discussed in section 1.

To this end, we follow the experimental setup of the most recent work, TCNLM and *quantitatively* compare the performance of our models (i.e., ctx-DocNADE and ctx-DocNADEe) in terms of topic coherence (NPMI) on BNC dataset. Table 7 (left) shows NPMI scores of different models, where the results suggest that our contribution (i.e., ctx-DocNADE) of introducing language concepts into BoW topic model (i.e., DocNADE) improves topic coherence[5]. The better performance for high val-

---

[3] https://github.com/jhlau/topically-driven-language-model

[4] PPL scores in (RV/FV) settings: DocNADE (665/1375) outperforms ProdLDA (1168/2097) on 200 topics

[5] NPMI over (50/200) topics learned on 20NS by: ProdLDA (.24/.19) and DocNADE (.15/.12) in RV setting

| Model | Coherence (NPMI) | | |
|---|---|---|---|
| | 50 | 100 | 150 |
| *(sliding window=20)* | | | |
| LDA# | .106 | .119 | .119 |
| NTM# | .081 | .070 | .072 |
| TDLM(s)# | .102 | .106 | .100 |
| TDLM(l)# | .095 | .101 | .104 |
| Topic-RNN(s)# | .102 | .108 | .102 |
| Topic-RNN(l)# | .100 | .105 | .097 |
| TCNLM(s)# | .114 | .111 | .107 |
| TCNLM(l)# | .101 | .104 | .102 |
| DocNADE | .097 | .095 | .097 |
| ctx-DocNADE*($\lambda$=0.2) | .102 | .103 | .102 |
| ctx-DocNADE*($\lambda$=0.8) | .106 | .105 | .104 |
| ctx-DocNADEe*($\lambda$=0.2) | .098 | .101 | - |
| ctx-DocNADEe*($\lambda$=0.8) | .105 | .104 | - |
| *(sliding window=110)* | | | |
| DocNADE | .133 | .131 | .132 |
| ctx-DocNADE*($\lambda$=0.2) | .134 | .141 | .138 |
| ctx-DocNADE*($\lambda$=0.8) | .139 | .142 | .140 |
| ctx-DocNADEe*($\lambda$=0.2) | .133 | .139 | - |
| ctx-DocNADEe*($\lambda$=0.8) | .135 | .141 | - |

| Topic | Model | Topic-words (ranked by their probabilities in topic) |
|---|---|---|
| environment | TCNLM# | pollution, emissions, nuclear, waste, environmental |
| | ctx-DocNADE* | ozone, pollution, emissions, warming, waste |
| | ctx-DocNADEe* | pollution, emissions, dioxide, warming, environmental |
| politics | TCNLM# | elections, economic, minister, political, democratic |
| | ctx-DocNADE* | elections, democracy, votes, democratic, communist |
| | ctx-DocNADEe* | democrat, candidate, voters, democrats, poll |
| art | TCNLM# | album, band, guitar, music, film |
| | ctx-DocNADE* | guitar, album, band, bass, tone |
| | ctx-DocNADEe* | guitar, album, pop, guitars, song |
| facilities | TCNLM# | bedrooms, hotel, garden, situated, rooms |
| | ctx-DocNADE* | bedrooms, queen, hotel, situated, furnished |
| | ctx-DocNADEe* | hotel, bedrooms, golf, resorts, relax |
| business | TCNLM# | corp, turnover, unix, net, profits |
| | ctx-DocNADE* | shares, dividend, shareholders, stock, profits |
| | ctx-DocNADEe* | profits, growing, net, earnings, turnover |
| expression | TCNLM# | eye, looked, hair, lips, stared |
| | ctx-DocNADE* | nodded, shook, looked, smiled, stared |
| | ctx-DocNADEe* | charming, smiled, nodded, dressed, eyes |
| education | TCNLM# | courses, training, students, medau, education |
| | ctx-DocNADE* | teachers, curriculum, workshops, learning, medau |
| | ctx-DocNADEe* | medau, pupils, teachers, schools, curriculum |

**Table 7:** (Left): Topic coherence (NMPI) scores of different models for 50, 100 and 150 topics on BNC dataset. The *sliding window* is one of the hyper-parameters for computing topic coherence (Röder et al., 2015; Wang et al., 2018). A *sliding window* of 20 is used in TCNLM; in addition we also present results for a window of size 110. $\lambda$ is the mixture weight of the LM component in the topic modeling process, and (s) and (l) indicate small and large model, respectively. The symbol '-' indicates no result, since word embeddings of 150 dimensions are not available from glove vectors. (Right): The top 5 words of seven learnt topics from our models and TCNLM. The asterisk (*) indicates our proposed models and (#) taken from TCNLM (Wang et al., 2018).

ues of $\lambda$ illustrates the relevance of the LM component for topic coherence (DocNADE corresponds to $\lambda$=0). Similarly, the inclusion of word embeddings (i.e., ctx-DocNADEe) results in more coherent topics than the baseline DocNADE. Importantly, while ctx-DocNADEe is motivated by sparse data settings, the BNC dataset is neither a collection of short-text nor a corpus of few documents. Consequently, ctx-DocNADEe does not show improvements in topic coherence over ctx-DocNADE.

In Table 7 (right), we further *qualitatively* show the top 5 words of seven topics (topic name summarized by Wang et al. (2018)) from TCNML and our models. Observe that ctx-DocNADE captures a topic *expression* that is a collection of only verbs in the past participle. Since the BNC dataset is unlabeled, we are here restricted to comparing model performance in terms of topic coherence only.

### 3.3 APPLICABILITY: TEXT RETRIEVAL AND CATEGORIZATION

*Text Retrieval*: We perform a document retrieval task using the short-text and long-text documents with label information. We follow the experimental setup similar to Lauly et al. (2017), where all test documents are treated as queries to retrieve a fraction of the closest documents in the original training set using cosine similarity measure between their `textTOvec` representations (section 2.2). To compute retrieval precision for each fraction (e.g., 0.0001, 0.005, 0.01, 0.02, 0.05, etc.), we average the number of retrieved training documents with the same label as the query. For multi-label datasets, we average the precision scores over multiple labels for each query. Since, Salakhutdinov & Hinton (2009) and Lauly et al. (2017) have shown that RSM and DocNADE strictly outperform LDA on this task, we solely compare DocNADE with our proposed extensions.

Table 3 and 4 show the retrieval precision scores for the short-text and long-text datasets, respectively at retrieval fraction 0.02. Observe that the introduction of both pre-trained embeddings and language/contextual information leads to improved performance on the IR task noticeably for short texts. We also investigate topic modeling without pre-processing and filtering certain words, i.e. the FV setting and find that the DocNADE(FV) or glove(FV) improves IR precision over the baseline RV setting. Therefore, we opt for the FV in the proposed extensions. On an average over the 8 short-text and 6 long-text datasets, *ctx-DocNADEe* reports a gain of 7.1% (.630 vs .588) (Table 3) 6.0% (.601 vs .567) (Table 4), respectively in precision compared to DocNADE(RV). To further compare with TDLM, our proposed models (ctx-DocNADE and ctx-DocNADEe) outperform it by a notable margin for all the short-text datasets, i.e., a gain of 14.5% (.630 vs .550: ctx-DocNADEe vs TDLM)

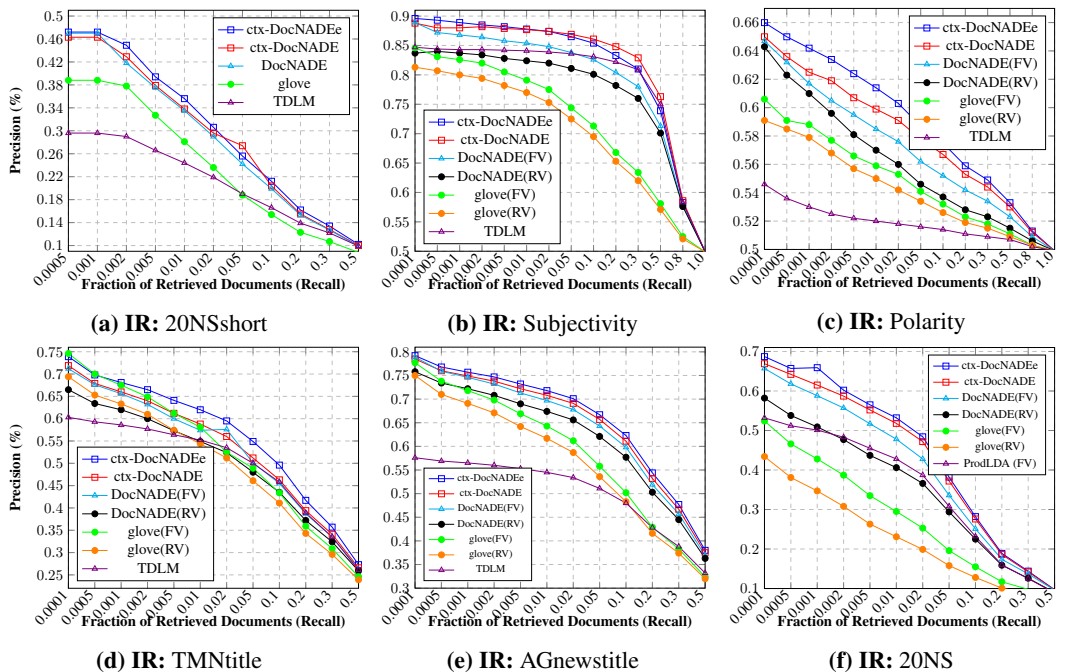

**Figure 3:** Retrieval performance (IR-precision) on 6 datasets at different fractions

in IR-precision. In addition, the deep variant ($d$=3) with embeddings, i.e., ctx-DeepDNEe shows competitive performance on TREC6 and Subjectivity datasets.

Figures (3a, 3b, 3c, 3d, 3e and 3f) illustrate the average precision for the retrieval task on 6 datasets. Observe that the ctx-DocNADEe outperforms DocNADE(RV) at all the fractions and demonstrates a gain of 6.5% (.615 vs .577) in precision at fraction 0.02, averaged over 14 datasets. Additionally, our proposed models outperform TDLM and ProdLDA[6] (for 20NS) by noticeable margins.

*Text Categorization*: We perform text categorization to measure the quality of our `textTovec` representations. We consider the same experimental setup as in the retrieval task and extract `textTOvec` of 200 dimension for each document, learned during the training of ctx-DocNADE variants. To perform text categorization, we employ a logistic regression classifier with $L2$ regularization. While, *ctx-DocNADEe* and *ctx-DeepDNEe* make use of glove embeddings, they are evaluated against the topic model baselines with embeddings. For the short texts (Table 3), the *glove* leads DocNADE in classification performance, suggesting a need for distributional priors in the topic model. Therefore, the ctx-DocNADEe reports a gain of 4.8% (.705 vs .673) and 3.6% (.618 vs .596) in $F1$, compared to DocNADE(RV) on an average over the short (Table 3) and long (Table 4) texts, respectively. In result, a gain of 4.4% (.662 vs .634) overall.

In terms of classification accuracy on 20NS dataset, the scores are: DocNADE (0.734), ctx-DocNADE (0.744), ctx-DocNADEe (0.751), NTM (0.72) and SCHOLAR (0.71). While, our proposed models, i.e., ctx-DocNADE and ctx-DocNADEe outperform both NTM (results taken from Cao et al. (2015), Figure 2) and SCHOLAR (results taken from Card et al. (2017), Table 2), the DocNADE establishes itself as a strong neural topic model baseline.

### 3.4 INSPECTION OF LEARNED REPRESENTATIONS

To further interpret the topic models, we analyze the meaningful semantics captured via topic extraction. Table 8 shows a topic extracted using 20NS dataset that could be interpreted as *computers*, which are (sub)categories in the data, confirming that meaningful topics are captured. Observe that

---

[6] IR-precision scores at 0.02 retrieval fraction on the short-text datasets by ProdLDA: *20NSshort* (.08), *TREC6* (.24), *R21578title* (.31), *Subjectivity* (.63) and *Polarity* (.51). Therefore, the *DocNADE, ctx-DocNADE and ctx-DocNADEe outperform ProdLDA in both the settings*: data sparsity and sufficient co-occurrences.

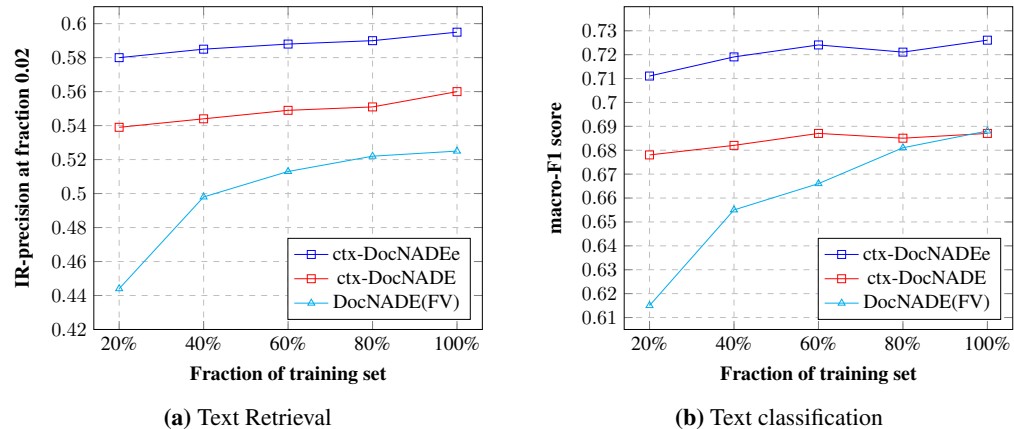

**(a)** Text Retrieval  **(b)** Text classification

**Figure 4:** Evaluations at different fractions (20%, 40%, 60%, 80%, 100%) of the training set of TMNtitle

the ctx-DocNADEe extracts a more coherent topic due to embedding priors. To *qualitatively* inspect the contribution of word embeddings and `textTOvec` representations in topic models, we analyse the text retrieved for each query using the representations learned from DocNADE and ctx-DoocNADEe models. Table 9 illustrates the retrieval of the top 3 texts for an input query, selected from *TMNtitle* dataset, where #match is YES if the query and retrievals have the same class label. Observe that ctx-DocNADEe retrieves the top 3 texts, each with no unigram overlap with the query.

| DocNADE | ctx-DocNADE | ctx-DocNADEe |
|---|---|---|
| vga, screen, | computer, color, | svga, graphics |
| computer, sell, | screen, offer, | bar, macintosh, |
| color, powerbook, | vga, card, | san, windows, |
| sold, cars, | terminal, forsale, | utility, monitor, |
| svga, offer | gov, vesa | computer, processor |
| .554 | .624 | .667 |

| | *Query* :: "emerging economies move ahead nuclear plans" | #match |
|---|---|---|
| ctx--DocNADEe | #IR1 :: imf sign lifting japan yen | YES |
| | #IR2 :: japan recovery takes hold debt downgrade looms | YES |
| | #IR3 :: japan ministers confident treasuries move | YES |
| DocNADE | #IR1 :: nuclear regulator back power plans | NO |
| | #IR2 :: defiant iran plans big rise nuclear | NO |
| | #IR3 :: japan banks billion nuclear operator sources | YES |

**Table 8:** A topic of 20NS dataset with coherence  **Table 9:** Illustration of the top-3 retrievals for an input query

Additionally, we show the quality of representations learned at different fractions (20%, 40%, 60%, 80%, 100%) of training set from TMNtitle data and use the same experimental setup for the IR and classification tasks, as in section 3.3. In Figure 4, we quantify the quality of representations learned and demonstrate improvements due to the proposed models, i.e., ctx-DocNADE and ctx-DocNADEe over DocNADE at different fractions of the training data. Observe that the gains in both the tasks are large for smaller fractions of the datasets. For instance, one of the proposed models, i.e., ctx-DocNADEe (vs DocNADE) reports: (1) a precision (at 0.02 fraction) of 0.580 vs 0.444 at 20% and 0.595 vs 0.525 at 100% of the training set, and (2) an F1 of 0.711 vs 0.615 at 20% and 0.726 vs 0.688 at 100% of the training set. Therefore, the findings conform to our second contribution of improving topic models with word embeddings, especially in the sparse data setting.

## 3.5 CONCLUSION

In this work, we have shown that accounting for language concepts such as word ordering, syntactic and semantic information in neural autoregressive topic models helps to better estimate the probability of a word in a given context. To this end, we have combined a neural autoregressive topic- (i.e., DocNADE) and a neural language (e.g., LSTM-LM) model in a single probabilistic framework with an aim to introduce language concepts in each of the autoregressive steps of the topic model. This facilitates learning a latent representation from the entire document whilst accounting for the local dynamics of the collocation patterns, encoded in the internal states of LSTM-LM. We further augment this complementary learning with external knowledge by introducing word embeddings. Our experimental results show that our proposed modeling approaches consistently outperform state-of-the-art generative topic models, quantified by generalization (perplexity), topic interpretability (coherence), and applicability (text retrieval and categorization) on 15 datasets.

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

## A DATA DESCRIPTION

We use 14 different datasets: (1) 20NSshort: We take documents from 20NewsGroups data, with document size less (in terms of number of words) than 20. (2) TREC6: a set of questions (3) Reuters21578title: a collection of new stories from nltk.corpus. We take titles of the documents. (4) Subjectivity: sentiment analysis data. (5) Polarity: a collection of positive and negative snippets acquired from Rotten Tomatoes (6) TMNtitle: Titles of the Tag My News (TMN) news dataset. (7) AGnewstitle: Titles of the AGnews dataset. (8) Reuters8: a collection of news stories, processed and released by (9) Reuters21578: a collection of new stories from nltk.corpus. (10) 20NewsGroups: a collection of news stories from nltk.corpus. (11) RCV1V2 (Reuters): www.ai.mit.edu/projects/jmlr/papers/volume5/lewis04a/lyrl2004_rcv1v2_README.htm (12) 20NSsmall: We sample 20 document for training from each class of the 20NS dataset. For validation and test, 10 document for each class. (13) TMN: The Tag My News (TMN) news dataset. (14) Sixxx Requirement OBjects (SiROBs): a collection of paragraphs extracted from industrial tender documents (our industrial corpus).

| **Label:** *training* |
|---|
| Instructors shall have tertiary education and experience in the operation and maintenance of the equipment or sub-system of Plant. They shall be proficient in the use of the English language both written and oral. They shall be able to deliver instructions clearly and systematically. The curriculum vitae of the instructors shall be submitted for acceptance by the Engineer at least 8 weeks before the commencement of any training. |
| **Label:** *maintenance* |
| The Contractor shall provide experienced staff for 24 hours per Day, 7 Days per week, throughout the Year, for call out to carry out On-call Maintenance for the Signalling System. |
| **Label:** *cables* |
| Unless otherwise specified, this standard is applicable to all cables which include single and multi-core cables and wires, Local Area Network (LAN) cables and Fibre Optic (FO) cables. |
| **Label:** *installation* |
| The Contractor shall provide and permanently install the asset labels onto all equipment supplied under this Contract. The Contractor shall liaise and co-ordinate with the Engineer for the format and the content of the labels. The Contractor shall submit the final format and size of the labels as well as the installation layout of the labels on the respective equipment, to the Engineer for acceptance. |
| **Label:** *operations, interlocking* |
| It shall be possible to switch any station Interlocking capable of reversing the service into "Auto-Turnaround Operation". This facility once selected shall automatically route Trains into and out of these stations, independently of the ATS system. At stations where multiple platforms can be used to reverse the service it shall be possible to select one or both platforms for the service reversal. |

**Table 10:** SiROBs data: Example Documents (Requirement Objects) with their types (label).

| **Hyperparameter** | **Search Space** |
|---|---|
| learning rate | [0.001] |
| hidden units | [200] |
| iterations | [2000] |
| activation function | sigmoid |
| $\lambda$ | [1.0, 0.8, 0.5, 0.3, 0.1, 0.01, 0.001] |

**Table 11:** Hyperparameters in Generalization in the DocNADE and ctx-DocNADE variants for 200 topics

The SiROBs is our industrial corpus, extracted from industrial tender documents. The documents contain requirement specifications for an industrial project for example, *railway metro construction*. There are 22 types of requirements i.e. class labels (multi-class), where a requirement is a paragraph or collection of paragraphs within a document. We name the requirement as Requirement Objects (ROBs). Some of the requirement types are *project management*, *testing*, *legal*, *risk analysis*, *financial cost*, *technical requirement*, etc. We need to classify the requirements in the tender documents and assign each ROB to a relevant department(s). Therefore, we analyze such documents to automate decision making, tender comparison, similar tender as well as ROB retrieval and assigning ROBs to a relevant department(s) to optimize/expedite tender analysis. See some examples of ROBs from SiROBs corpus in Table 10.

## B    EXPERIMENTAL SETUP

### B.1    EXPERIMENTAL SETUP AND HYPERPARAMETERS FOR GENERALIZATION TASK

See Table 11 for hyperparameters used in generalization.

### B.2    EXPERIMENTAL SETUP AND HYPERPARAMETERS FOR IR TASK

We set the maximum number of training passes to 1000, topics to 200 and the learning rate to 0.001 with $tanh$ hidden activation. For model selection, we used the validation set as the query set and used the average precision at 0.02 retrieved documents as the performance measure. Note that the

| Hyperparameter | Search Space |
|---|---|
| retrieval fraction | [0.02] |
| learning rate | [0.001] |
| hidden units | [200] |
| activation function | tanh |
| iterations | [2000] |
| $\lambda$ | [1.0, 0.8, 0.5, 0.3, 0.1, 0.01, 0.001] |

**Table 12:** Hyperparameters in the Document Retrieval task.

| Dataset | Model | $\lambda$ | | |
|---|---|---|---|---|
| | | 1.0 | 0.1 | 0.01 |
| 20NSshort | ctx-DocNADE | 899.04 | 829.5 | 842.1 |
| | ctx-DocNADEe | 890.3 | 828.8 | 832.4 |
| Subjectivity | ctx-DocNADE | 982.8 | 977.8 | 966.5 |
| | ctx-DocNADEe | 977.1 | 975.0 | 964.2 |
| TMNtitle | ctx-DocNADE | 1898.1 | 1482.7 | 1487.1 |
| | ctx-DocNADEe | 1877.7 | 1480.2 | 1484.7 |
| AGnewstitle | ctx-DocNADE | 1296.1 | 861.1 | 865 |
| | ctx-DocNADEe | 1279.2 | 853.3 | 862.9 |
| Reuters-8 | ctx-DocNADE | 336.1 | 313.2 | 311.9 |
| | ctx-DocNADEe | 323.3 | 312.0 | 310.2 |
| 20NS | ctx-DocNADE | 1282.1 | 1209.3 | 1207.2 |
| | ctx-DocNADEe | 1247.1 | 1211.6 | 1206.1 |

**Table 13:** Perplexity scores for different $\lambda$ in Generalization task: Ablation over validation set

labels are not used during training. The class labels are only used to check if the retrieved documents have the same class label as the query document. To perform document retrieval, we use the same train/development/test split of documents discussed in data statistics (experimental section) for all the datasets during learning.

See Table 12 for the hyperparameters in the document retrieval task.

## B.3 EXPERIMENTAL SETUP FOR DOC2VEC MODEL

We used gensim (`https://github.com/RaRe-Technologies/gensim`) to train Doc2Vec models for 12 datasets. Models were trained with distributed bag of words, for 1000 iterations using a window size of 5 and a vector size of 500.

## B.4 CLASSIFICATION TASK

We used the same split in training/development/test as for training the Doc2Vec models (also same split as in IR task) and trained a regularized logistic regression classifier on the inferred document vectors to predict class labels. In the case of multilabel datasets (`R21578,R21578title, RCV1V2`), we used a one-vs-all approach. Models were trained with a liblinear solver using L2 regularization and accuracy and macro-averaged F1 score were computed on the test set to quantify predictive power.

## B.5 EXPERIMENTAL SETUP FOR GLOVE-DMM AND GLOVE-LDA MODELS

We used LFTM (`https://github.com/datquocnguyen/LFTM`) to train glove-DMM and glove-LDA models. Models were trained for 200 iterations with 2000 initial iterations using 200 topics. For short texts we set the hyperparameter beta to 0.1, for long texts to 0.01; the mixture parameter lambda was set to 0.6 for all datasets. The setup for the classification task was the same as for doc2vec; classification was performed using relative topic proportions as input (i.e. we inferred the topic distribution of the training and test documents and used the relative distribution as input

| Dataset | Model | $\lambda$ | | | |
|---|---|---|---|---|---|
| | | 1.0 | 0.8 | 0.5 | 0.3 |
| 20NSshort | ctx-DocNADE | 0.264 | 0.265 | 0.265 | 0.265 |
| | ctx-DocNADEe | 0.277 | 0.277 | 0.278 | 0.276 |
| Subjectivity | ctx-DocNADE | 0.874 | 0.874 | 0.873 | 0.874 |
| | ctx-DocNADEe | 0.868 | 0.868 | 0.874 | 0.87 |
| Polarity | ctx-DocNADE | 0.587 | 0.588 | 0.591 | 0.587 |
| | ctx-DocNADEe | 0.602 | 0.603 | 0.601 | 0.599 |
| TMNtitle | ctx-DocNADE | 0.556 | 0.557 | 0.559 | 0.568 |
| | ctx-DocNADEe | 0.604 | 0.604 | 0.6 | 0.6 |
| TMN | ctx-DocNADE | 0.683 | 0.689 | 0.692 | 0.694 |
| | ctx-DocNADEe | 0.696 | 0.698 | 0.698 | 0.7 |
| AGnewstitle | ctx-DocNADE | 0.665 | 0.668 | 0.678 | 0.689 |
| | ctx-DocNADEe | 0.686 | 0.688 | 0.695 | 0.696 |
| 20NSsmall | ctx-DocNADE | 0.352 | 0.356 | 0.366 | 0.37 |
| | ctx-DocNADEe | 0.381 | 0.381 | 0.375 | 0.353 |
| Reuters-8 | ctx-DocNADE | 0.863 | 0.866 | 0.87 | 0.87 |
| | ctx-DocNADEe | 0.875 | 0.872 | 0.873 | 0.872 |
| 20NS | ctx-DocNADE | 0.503 | 0.506 | 0.513 | 0.512 |
| | ctx-DocNADEe | 0.524 | 0.521 | 0.518 | 0.511 |
| R21578 | ctx-DocNADE | 0.714 | 0.714 | 0.714 | 0.714 |
| | ctx-DocNADEe | 0.715 | 0.715 | 0.715 | 0.714 |
| SiROBs | ctx-DocNADE | 0.409 | 0.409 | 0.408 | 0.408 |
| | ctx-DocNADEe | 0.41 | 0.411 | 0.411 | 0.409 |
| AGnews | ctx-DocNADE | 0.786 | 0.789 | 0.792 | 0.797 |
| | ctx-DocNADEe | 0.795 | 0.796 | 0.8 | 0.799 |

**Table 14:** $\lambda$ for IR task: Ablation over validation set at retrieval fraction 0.02

for the logistic regression classifier). Similarly, for the IR task, similarities were computed based on the inferred relative topic distribution.

### B.6 EXPERIMENTAL SETUP FOR PRODLDA

We run ProdLDA (`https://github.com/akashgit/autoencoding_vi_for_topic_models`) on the short-text datasets in the FV setting to generate document vectors for IR-task. We use 200 topics for a fair comparison with other baselines used for the IR tasks. We infer topic distribution of the training and test documents and used the relative distribution as input for the IR task, similar to section 3.3.

To fairly compare PPL scores of ProdLDA and DocNADE in the RV setting, we take the pre-processed 20NS dataset released by ProdLDA and run DocNADE for 200 topics. To further compare them in the FV setting, we run ProdLDA (`https://github.com/akashgit/autoencoding_vi_for_topic_models`) on the processed 20NS dataset for 200 topics used in this paper.

## C  ABLATION OVER THE MIXTURE WEIGHT $\lambda$

### C.1  $\lambda$ FOR GENERALIZATION TASK

See Table 13.

### C.2  $\lambda$ FOR IR TASK

See Table 14.

# D ADDITIONAL BASELINES

## D.1 DOCNADE VS SCHOLAR

*PPL scores over 20 topics*: DocNADE (752) and SCHOLAR (921), i.e., DocNADE outperforms SCHOLAR in terms of generalization.

*Topic coherence (NPMI) using 20 topics*: DocNADE (.18) and SCHOLAR (.35), i.e., SCHOLAR (Card et al., 2017) generates more coherence topics than DocNADE, though worse in PPL and text classification (see section 3.3) than DocNADE, ctx-DocNADE and ctx-DocNADEe.

*IR tasks*: Since, SCHOLAR (Card et al., 2017) without meta-data equates to ProdLDA and we have shown in section 3.3 that ProdLDA is worse on IR tasks than our proposed models, therefore one can infer the performance of SCHOLAR on IR task.

The experimental results above suggest that the DocNADE is better than SCHOLAR in generating good representations for downstream tasks such as information retrieval or classification, however falls behind SCHOLAR in interpretability. The investigation opens up an interesting direction for future research.

