# OpenReview forum: "textTOvec: DEEP CONTEXTUALIZED NEURAL AUTOREGRESSIVE TOPIC MODELS OF LANGUAGE WITH DISTRIBUTED COMPOSITIONAL PRIOR"
_ICLR.cc/2019/Conference_

### Official Review · AnonReviewer2 · 2018-10-26
**Extends existing DocNADE model by replacing the feedforward network with an LSTM**

**Rating:** 6
**Confidence:** 4

**Review:**

The paper extends an existing topic model - DocNADE - by replacing the feedforward part of the network which combines the textual context with an LSTM sequence model. Hence this paper fits in a long tradition of work which aims to extend the bad-of-words model from the original LDA paper with some sequence information.

The authors do a commendable job in thoroughly evaluating the proposed extension, using a number of evaluations based on perplexity, topic coherence, and text retrieval and categorization.

My main problem with the paper as it stands is that it a) arguably oversells the contribution, and b) is unclear when explaining certain crucial aspects of the model.

It would also help to have a clearer statement of whether the contribution here is on the document modeling side, or the language modeling side. Motivation is provided from both angles, but the evaluation focuses largely on the topic modeling (which is fine, just need to say it).

More specific comments:
--
The abstract should mention that the DocNADE model already exists, and that the contribution of the current work is to extend that existing model in a particular way. For those readers unfamiliar with DocNADE, this will help situate the work with regard to the existing literature.

Using existing word embeddings as a 'prior' for the LSTM word embeddings is a completely standard alternative now to learning those embeddings from scratch. I'm not sure that can count as a second, major contribution of the paper. (I'm also not sure that either extension to DocNADE warrants a new name, but I'll leave that to the authors' judgement.)

I'm confused by one aspect of the DocNADE model: "the topic assigned ...equally depends all the other words appearing in the same document". But the model is generative, no? And eqn 1 suggests that each word is generated conditioned on the *previous* words in the document, or did I miss something?

Related to this point, DocNADE transforms its BoWs into a sequence. But what's the order? Is it just the order of the words in the document? In which case it's very similar to the LSTM extension, except the LSTM keeps the order in the history, whereas the bag-of-words model doesn't.

Relation to generative models: LDA is a generative model with a generative story. It's not completely obvious to me what the generative story is in the new model. Talking about "distributed compositional priors" doesn't help, since I'm assuming these aren't priors in a Bayesian modeling sense? (It's also not clear in what sense these "priors" are compositional, but that's a separate question.)

Equation 2: what's the motivation for mixing the LSTM history with the bag-of-words (esp. if the history is from the same bag of words in each case). Why not just use the LSTM?

It would be useful to state in the main body of the text what the value of lambda ends up being. In 3.1 there's a suggestion this might be 0.01, but that effectively ignores the LSTM?

Similar question: how can the DocNADE model provide a *global* context if the model is generative?

Perplexity is a reasonable thing to measure, but presumably the auto-regressive nature of the LSTM means that it's more-or-less guaranteed to do better than a bag-of-words model? I wonder if it's worth acknowledging this fact?

I don't understand why lambda has to be zero "to compute the exact log-likelihood".

The first line of the conclusion doesn't say much: it's pretty obvious that the ordering of the words is going to help better estimate the probability of a word in a given context; 50 years of language modeling research has already taught us that.

Minor presentational comments:
--
Some of the hyphenation looks odd, eg in the title. Are you using the standard LaTeX hyphenation settings?

Strictly speaking, I'm not sure that 'bear' in the example is a proper noun.

"orderless sets of words": bags, not sets, since the counts matter, no?

The tables are too small, with a lot of numbers in them. One option is to move some of the details to the Appendix. Either way there needs to be more summary in the main body explaining what the numbers tell us.

---

> ### Author Response · Authors · 2018-11-20
> **Clarification about the queries/comments**
>
> Thanks for your review comments and positive feedback about evaluation: "commendable job in thoroughly evaluating".
>
> Please, see Table 7, Table 3, Figure 3 and Figure 4 in the revised version for additional experimental results.
>
> "explaining certain crucial aspects":
> We have tried to better motivate our tasks/contributions in the revised version, along with recent studies (including the suggested baselines) in the introduction and baseline sections. We would appreciate if could specifically point out the crucial aspects, in case we missed any.
>
> "clearer statement... which is fine, just need to say it":
> We state several times throughout the paper (e.g., in the abstract, contribution1, contribution2, etc.) that we focus on improving topic models using language concepts learned from language models. Based on your feedback, we now also emphasize this in the evaluation section (1st paragraph) and also updated the title.
>
> "embeddings in LSTM":
> We clarify that our contribiution2 is about incorporating word embeddings into topic models to handle data sparsity challenges in topic models. To this end, we incorporate word embeddings into topic models via an LSTM and compare our contribution, i.e., ctx-DocNADEe with related works, such as glove-LDA, glove-DMM and gaussian-LDA that also extend topic models with word embedding features. Our modeling approach offers an easy integration of word embeddings into topic models, that substantially improves topic coherence.
>
> "DocNADE: generative and global semantics":
> DocNADE is a generative model with orderless BoW input.  As detailed in equation 12 (Larochelle and Lauly, 2012), a random permutation of its words is used to produce the observed document - this random order is re-shuffled during learning so that all information related to the original word order is lost.
> The order of the input in DocNADE is arbitary, while the original order of words in the document is presented via LSTM. Our contribution of combining the two networks is further motivated by recent studies, such as ELMo (Peters et al., 2018) that have shown that internal states of LSTM-LM capture language concepts (such as word order, syntactic and semantic information). Therefore, we have extended DocNADE by introducing these language concepts and shown improvements on different tasks using 15 datasets.
>
> "Prior and compositional":
> The term 'prior' is used for the external information i.e., word embeddings encoding word relatedness in distributional semantic space. The term 'compositional' is used for the compositional property of RNN-LSTM, where LSTM-LM generates latent vectors (i.e., internal states of LSTM) for each input context via sequential compositionality. Each of the internal states for the corresponding word sequence captures language concepts that are presented to DocNADE, correspondingly at each of the autoregressive steps.
>
> "history":
> The two history vectors: v<i and c_i are two different representations, in the sense that v<i is sampled from orderless BoW. This arbitary sequence is different to c_i in the sense that c_i is the original sequence of preceding words for the ith word in the document. The motivation is to combine the benefits of the two networks (i.e., DocNADE and LSTM-LM) to improve topic models, i.e., introduce local dynamics into global semantics of topic models. We control the mixture via a mixture weight $\lambda$.
>
> "lambda values":
> It is a hyperparamter, determined using the validation set; optimal values vary between datasets. Please see appendices (Table 13) for the ablation study.
>
> "perplexity measure":
> As discussed, we set lambda to 0 during evaluation, i.e., no LSTM component is used during testing. We only exploit LSTM-LM during training to learn language concepts, encoded in the W matrix. Beyond perplexity, we also perform topic coherence, text retrieval and classification tasks.
>
> "pseudo log likelihood":
> p(v) is exact for DocNADE (see details in Larochelle and Lauly (2012)), however not in its extensions ctx-DocNADE and ctx-DocNADEe. While in the proposed models (as discussed in section 2.2), each autoregressive conditional p(v_i | v<i) is a function of h_i and h_i (Table 1) is a function of v<i and context c_i, the likelihood is not exact due to difference in contexts: v<i and c_i. In further detail, v<i is based on the orderless BoW (and therefore an arbitary order) where each v_i is the index in the vocabulary. In contrast, c_i is the context of ith word in the document (and therefore the original word sequence of the context preceding the ith word). Importantly, to compute PPL scores during testing, we set lambda to 0, allowing a fair comparison between DocNADE and our proposed models (that is, the test liklihood is exact again).
>
> "first line of conclusion":
> We have updated the conclusion section.
>
> "using the standard LaTeX hyphenation settings?":
> Yes.
>
> "bear as proper noun":
> We use stanford parser (http://nlp.stanford.edu:8080/parser/index.jsp) to perform tagging.

---

### Official Review · AnonReviewer3 · 2018-11-02
**related work not great, but experiements extensive otherwise**

**Rating:** 8
**Confidence:** 4

**Review:**

DocNADE has great performance so this is a welcome bit of
research extending it.

There has been a huge amount of activity in combining topic models with
(1) embeddings and (2) neural networks such as LSTMs and RNNs.
I will say I have great sympathy for the poor author trying to do
fair comparisons against start-of-the-art because the standards are
moving quickly.

In this case, some neural network papers I have seen are TopicRNNs by
Dieng, Wang Gao and Paisley, and LLA by Zaheer, Ahmed and Smola.  The
latter is still a bag-of-words model and but places the LSTM over the
sequence of topic proportions.  The Gauss-LDA and glove-DMM work is
fairly dated (in our fast-paced ML world) and their performance is
known to be poor, as some papers in 2017 show.  Now I know historically LDA has been fairly poor
with IR tasks, but I would expect the recent supervised LDA methods,
some also have word embeddings, to do better as well.
So the discussion of related work and comparative experiments
are poor.

If you want to illustrated good improvememts gained using embeddings,
it helps to try different proportions, say 20/40/60/80% of a data
set and plot.  Usually, you should see embeddings aid performance
dramatically for smaller fractions of data sets.  Hence, your results
seem strange.

Note the data sets are all fairly small, which makes me wonder about
the computation time.  Could you give some computational performance
stats for a data set?

In section 2.2 top of page 5, why is it "pseudo" log likelihood?
Isn't that formula exact?

The paper has a relatively small part devoted to the model, and
virtually nothing on the algorithm, although this is probably covered
in earlier DocNADE papers.   I'm assuming the model is
trained by SGD on the log likelihood with all the parameters
shoved in there in one go.  Is that right?  Would be nice to mention
whatever it is.

The use of four different kinds of evaluations (classification, IR,
perplexity, etc.) is good.  Note that the improvement over the earlier
DocNADE is quite small but clearly significant, and improvement of adding
embeddings seems even smaller, though seems better for short texts.
I wonder if the method for including embeddings is much good!
Not fully convinced.

AFTER RESPONSE:   Wow guys, what a great revision.  Thanks so much.

---

> ### Author Response · Authors · 2018-11-20
> **Improved related works, covered in total 13 related baselines (6 additional now in the revised version)**
>
> Thank you for your positive comments, especially about different kinds of evaluations and improvements.
>
> We agree that the fast-paced ML research in this area makes exhaustive comparisons challenging. However, we do attempt to cover the most recent, strong and comparable/fair baselines (within the limits on submission size) and have now included several additional baselines in the revised version. More specifically we have followed your suggestions and now include TDLM, Topic-RNN and TCNLM ("additional baselines", section 3.2) in the revised version (see also Table 3, Table 7, Figure 3 and Figure 4 in the revised version).
> However, most recent related studies focus on improving language models (LMs) using TMs: in contrast, the focus of our work is on improving TM for textual representations (short-text or long-text documents) by incorporating language concepts (e.g., word ordering, syntax, semantics, etc.) via neural LMs. In addition, we also address the challenges of topic learning that arise from limited context in collections of "short-text" and/or  "few" documents. Therefore, as part of our second contribution, we incorporate external knowledge, i.e., word embeddings into neural autoregressive TM to better model such document collections. It is worth noting that in contrast to this sparse data setting, the related approaches you mention were designed for collections of long-text as well as a sufficient number of documents.
>
> "Additional Baselines":
> We now compare our models to other approaches (TDLM, Topic-RNN and TCNLM) combining TMs with LM models. To this end, we follow the most recent approach presented in "Topic Compositional Neural Language Model" (Wang et al., 2018) and quantitatively compare TM performance in terms of topic coherence (NPMI) on the BNC dataset.
>
> *Additional experimental results on topic coherence* (comparing LDA, NTM, TDLM, Topic-RNN, TCNLM, DocNADE, ctx-DocNADE and ctx-DocNADEe on the BNC dataset) are included in the revised version (Table 7, Left and Right), showing competitive performance of our approach, although the related studies focus on improving topic models.
>
> **Additional experimental results on IR and classification tasks**: We run additional experiments and execute TDLM on all the short-text datasets to show its performance for the IR and classification tasks. Please see Table 3 in the revised version. In Figure 3, we have additionally included IR-precision by TDLM model at different fractions and show that our contributions (i.e., ctx-DocNADE and ctx-DocNADEe) outperform TDLM.
>
> Remark: Code for the other recent studies is not available.
>
> "embedding gain with different data fraction":
> As suggested, we have now included additional analysis to demonstrate embedding gains with different data fractions. Please, see section 3.4 and Figure 4.
>
> "pseudo log likelihood":
> p(v) is exact for DocNADE (see details in Larochelle and Lauly (2012)), however not in its extensions ctx-DocNADE and ctx-DocNADEe. While in the proposed models (as discussed in section 2.2), each autoregressive conditional p(v_i | v<i) is a function of h_i and h_i (Table 1) is a function of v<i and context c_i, the likelihood is not exact due to difference in contexts: v<i and c_i. In further detail, v<i is based on the orderless BoW (and therefore an arbitary order) where each v_i is the index in the vocabulary. In contrast, c_i is the context of ith word in the document (and therefore the original word sequence of the context preceeding the ith word). Importantly, to compute PPL scores during testing, we set lambda to 0, allowing a fair comparison between DocNADE and our proposed models (that is, the test liklihood is exact again).
>
> "trained by SGD on the log likelihood":
> Yes. We have now included such details in the revised version.
>
> "model description":
> We have devoted section 2.1, section 2.2, algorithm 1 and Table 1 to explain our proposed models as well as DocNADE in more detail.
>
> "performance on including embeddings":
> We have extended DocNADE models specifically to better model short-text or a corpus of few documents (see also "motivation2"); we have introduced word embeddings to better deal with this data sparsity and hence expect improvements resulting from embeddings mainly for such sparse data settings.
>
> For such short-text datasets, ctx-DocNADEe results in substantially improved IR, F1, PPL and topic coherence: (1) improved IR (DocNADE vs ctx-DocNADEe): 0.600 vs 0.630 in Table 3, (2) improved F1 (DocNADE vs ctx-DocNADEe): 0.683 vs 0.705 in Table 3, (3) improved PPL in Table 5 and (4) improved topic coherence (DocNADE vs ctx-DocNADEe): 0.755 vs 0.790. As suggested, we have now included additional analysis to demonstrate embedding gains with different data fractions. Please, see section 3.4 and Figure 4.
>
> Given these results on several datasets for the four tasks, we argue that the introduction of embeddings helps in improving topic models, especially in sparse data settings.

---

> > ### Author Response · Authors · 2018-11-24
> > **(continued) Response to additional queries (*please see revised version for improved related works*)**
> >
> > Please find the response below in continuation to the above one:
> >
> > About "computational performance":
> > -----------------------------------------------------
> > In terms of training time for 1 epoch, the ctx-DocNADE takes 4.40 and 109.12 seconds for 20NSshort (short-text documents) and 20NSsmall (long-text documents, but a corpus of few documents) datasets, respectively when run on a CPU machine and the number of threads is set to 1.  The batch size set to 100.
> >
> > About "recent supervised LDA methods":
> > ---------------------------------------------------------
> > In our work, we focus on unsupervised techniques in modeling text documents.
> >
> > About "comparative experiments":
> > -------------------------------------------------
> > We have now included in total 13 baselines (including 6 additional) in the revised version. Based on your feedback, we have also covered recent baselines, such as Topic-RNN, TCNLM, TDLM, etc. and compared with our proposed models for topic coherence and information retrieval (IR) tasks, where we have shown that our methods are competitive for topic coherence and outperform the related studies for IR by a large margin for 8 short-text datasets.
> >
> > **Please see the revised version for the improved related works and additional experiments.**

---

> ### Author Response · Authors · 2018-11-24
> **Thanks for revising rating to 'clear accept'**
>
> We are very happy for your comment about our revised version: "Wow guys, what a great revision".
>
> We appreciate and thank you for raising the overall rating.

---

### Official Review · AnonReviewer1 · 2018-11-02
**Method is not novel but results seem to be solid**

**Rating:** 7
**Confidence:** 4

**Review:**


Cons:
The proposed method is not novel. For example, Lauly et al., 2017 have proposed a similar way of combining LM and DocNADE. This paper does not provide some motivations or theories behind such artificial combination (i.e., just linearly combine their hidden state) to explain why it works better than other alternatives (e.g., what about adding some linear layers before combining h_i^{DN} and h_i^{LM}).

Pros:
However, the results seem to be solid and significantly better than the previous state-of-the-art methods. I think some recent neural topic models such as [1,2,3] are still missing even though there are already many tables in the paper (I am not an expert on neural topic modeling or embedding for IR tasks, so there might be others missing state of the arts which I am not aware of). In addition, why does Table 5 only compares perplexity between 3 methods and Table 6 only compares coherence between 4 or 5 methods, while there are 9 or 12 methods are compared in IR task (Table 3 and 4). What's the difficulty of comparing the coherence and perplexity of all different topic models (including [1,2,3])?
I will vote for acceptance if the mentioned baselines are also compared or there are good reasons why they cannot be compared.


Writing and presentation:
The quality of writing should be improved. Here are several examples.
1. In the abstract, the following sentence needs to be rewritten and the rule of capitalization should be consistent. "(2) Limited Context and/or Smaller training corpus of documents: In settings with a small number of word occurrences (i.e., lack of context) in short text or data sparsity in a corpus of few documents, the application of TMs is challenging."
2. I do not understand what's the purpose of the right figure in Figure 1. I think the paper does not do any matching like that.
3. In the 3rd paragraph of the introduction, "topmost" -> top most
4. The paper should have a related work section. In addition to the related work discussion scattered in the introduction, authors should discuss the difference between this work and Lauly et al., 2017. Authors should also include some related work such as [1,2,3].
5. Just below (1), "where," -> , where
6. In the last sentence of the paragraph after (1), you mentioned "v_{<i} are orderless", so what's the ordering used in experiments? Random ordering?
7. I guess "a" in algorithm 1 means sum_{k<i}(W_{:,v_k}), but I cannot find the explicit explanation about the purpose of "a".
8. For ctx-DocNADEe, is W+E the embedding of words at input layer in LM?
9. In the 3rd paragraph of section 2.2, you said: "each row vector W_{j,:} is a distribution over vocabulary of size K". Could W has negative values during optimization?  If yes, why a distribution representing a topic could have negative value. If no, you should explicitly mention this non-negativity constraint.
10. Why are some values in Table 12 and 13 missing?

[1] Cao, Z., Li, S., Liu, Y., Li, W., & Ji, H. (2015, January). A Novel Neural Topic Model and Its Supervised Extension. In AAAI (pp. 2210-2216).
[2] Srivastava, A., & Sutton, C. (2017). Autoencoding variational inference for topic models. ICLR
[3] Card, D., Tan, C., & Smith, N. A. (2017). A Neural Framework for Generalized Topic Models. arXiv preprint arXiv:1705.09296.

---

> ### Author Response · Authors · 2018-11-20
> **Clarification to minor comments/questions**
>
> "the purpose of the right figure in Figure 1":
> We illustrate the motivation for incorporating word embeddings into topic models.
>
> "a section about related works":
> We have now included the suggested related works in the revised version where we have motivated our task and contributions, for instance, "contribution1" and "contribution2" in the introduction section. DocNADE-LM (Lauly et al., 2017) is briefly mentioned in paragraph before the "contribution1". We have included [1]; however, as discussed above [2] and [3] have different motivation to our work, although a comparison is provided in the experimental section. Additionally, we have included more related works, such as TDLM, Topic-RNN and TCNLM.
>
> "linearly combine their hidden":
> ELMo (Peters et al., 2018), a recent study, demonstrated that the hidden/internal state of each word in LSTM-LM encodes language concepts such as word ordering, syntactic and semantic information. We employ these internal states to improve latent topic vectors, h^DN (equation 2). In our proposed modeling, we motivate the linear combination of h_i^{DN} and h_i^{LM} to maintain architectural simplicity, following Lauly et al. (2017). The motivation behind the combination is: For each word v_i of a document v, the hidden state h_{i}^{DN} at the ith autoregressive step encodes topic semantics via DocNADE, while h_{i}^{LM} encodes language concepts via LSTM-LM.
> However, further investigations about applying linear layers would be an interesting future activity.
>
> "ordering":
> Yes. Random ordering, following DocNADE.
>
> "purpose of "a"":
> It is a linear activation, mentioned on page 5 in paragraph before "ctx-DeepDNEe".
>
> "W+E the embedding of words at input layer in LM?":
> Yes, where W is trainable.
>
> "distribution representing a topic":
> Thanks for your insightful observation. Yes, we have rephrased it though it is the property of DocNADE. As mentioned in section 6.3 (of Larochelle and Lauly, 2012), the matrix W is being used to compute topics as well as word representations.
>
> "values in Table 12 and 13 of appendices":
> Given the extensive evaluation on 15 datasets in our work, we had to a run a large number of experiments (>400) and therefore tried to minimize the grid-search. We have included the missing numbers in the appendices of the revised version.

---

> ### Author Response · Authors · 2018-11-20
> **Clarification to additional queries**
>
> "Novelty":
> As mentioned by "AnonReviewer2" about our contributions, i.e., "This paper fits in a long tradition of work which aims to extend the bag-of-words model from the original LDA paper with some sequence information", we would like to restate the novelty/contribution of our work below:
> (1) We focus on improving topic models using language models within a unified neural network framework. Therefore, all evaluations are anchored on topic modeling.
> (2) To address challenges of topic modeling in sparse data settings we introduce word embeddings via LMs, thereby improving topic modeling for short-text or a corpus of few documents.
>
> "Motivation":
> As we have discussed in our submission: “recently, Peters et al. (2018) have shown that a deep contextualized LSTM-based language model (LSTM-LM) is able to capture different language concepts in a layer-wise fashion”. On other hand, the baseline DocNADE models word co-occurrences in orderless BoW fashion by a linear aggregation of word representations (equation 1), i.e., without incorporating the contextual information for *each word*.
>
> This has motivated us to introduce language concepts for each word (learned via LSTM-LM) into a autoregressive neural topic model (DocNADE). We jointly model the two neural networks in order to exploit the complementary learning and boost each of the autoregressive hidden representations (for every word, equation 2). This improves topic modeling *word-by-word* with contextualized word representations, word ordering information and local dynamics of the sequences.
>
> "DocNADE-LM":
> Unlike our contributions, DocNADE-LM (Lauly et a., 2017) restricts the history for each target word to the n − 1 preceding words, following an n-gram assumption, where the hidden states due to LM part are generated from a linear combination of n-1 word embeddings (not pre-trained). Additionally, DocNADE-LM is not extensively studied.
> Unlike DocNADE-LM, our contribution lies in marrying the two networks: DocNADE and LSTM-LM to incorporate language concepts for each word in the sequence in order to improve autoregressive topic models such as DocNADE.
>
> "Comparing coherence and perplexity of all different topic models":
> We have already compared our multi-fold contributions to several baselines (e.g., DocNADE, glove, glove-DMM, glove-LDA, gaussian-LDA, doc2vec, TDLM, Topic-RNN, TCNLM, etc.) that either (1) do not model contextualized word information (e.g., DocNADE, glove-DMM, glove-LDA and gaussian-LDA), or (2) do not incorporate word embeddings (e.g., DocNADE), or (3) ignore both in TMs (e.g., DocNADE) (4) incorporate word embeddings (e.g., glove-DMM, glove-LDA and gaussian-LDA) (5) jointly trained topic and language models (e.g., TDLM, Topic-RNN and TCNLM).
> Given the page limits, we have attempted to include related works for each of the different contributions within a single framework that we have proposed. One key benefit of our contributions is that it offers a general architecture addressing different challenges and extending state-of-the-art on different tasks, instead of having different models to deal with specific challenges/tasks.
>
> We have not included perplexity in Table 5 for glove-DMM, glove-LDA and Gaussian-LDA, since these baselines *ONLY* focus on improving topic models in terms of generating more coherent topics by incorporating word embeddings. Therefore, we compare them with our proposed models for topic coherence in Table 6. Additionally, we have already compared them for IR and categorization tasks. Since, Chang et al. (2009) and Newman et al. (2010) have shown that the perplexity is not a good metric for qualitative evaluation of topics, the practice of *introducing word embeddings into topic models* is often evaluated via topic coherence; in addition to this evaluation strategy, we also perform extrinsic evaluations via text retrieval and classification tasks.
>
> *Remark*: Since the baseline methods doc2vec and glove are not topic models but methods for generating paragraph/word embeddings, they are not part of Table 6. Instead, we have compared them with our proposed methods (Table 3 and 4) via text retrieval and categorization tasks to quantify the quality of representations learned.

---

> ### Author Response · Authors · 2018-11-20
> **"vote for acceptance if the mentioned baselines are also compared or reasoned"**
>
> Thank you for your insightful comments. We appreciate your positive comments: "solid and significantly better results".
>
> We have been running additional experiments for the suggested baselines and revised our submission based on your feedback.
>
> Following are the clarifications and answers to your questions:
>
> "I will vote for acceptance if the mentioned baselines are also compared or there are good reasons why they cannot be compared":
>
> As discussed in detail below, we have quantitatively *compared* our proposed models with *all* the suggested baselines [1,2,3] and also *reasoned* about different evaluations with glove-DMM, glove-LDA, etc., as suggested. Please see Table 7, Table 3, Figure 3 and section 3.3 (last paragraph) for additional experimental results/baselines in the revised version.
>
> In summary, we have now employed in total *13 baseline methods* (including 6 additional, as suggested by different reviewers): DocNADE, glove, glove-DMM, glove-LDA, gaussian-LDA, doc2vec, LDA, NTM[1], ProdLDA[2], SCHOLAR[3], TDLM, Topic-RNN and TCNLM to quantify our  contributions in different dimensions within a single model. In this work, we have focused on improving topic models (TM) and shown significantly better results in terms of perplexity, topic coherence, text retrieval and classification using 15 datasets.
> Given the limits on submission size and a very active area of research, we attempt to cover the most recent, strong and comparable/fair baselines in the revised version.
>
> "recent neural topic models" as suggested [1,2,3]:
>
> Our motivation in this work is to improve topic models with language concepts, learned via LSTM-LM.
> Since DocNADE (Lauly et a., 2017) outperforms LDA, RSM, o-RSM, autoencoders and word2vec baselines in terms of perplexity and information retrieval, we have chosen DocNADE  as baseline model that we extend (also acknowledged by "AnonReviewer3": "DocNADE has great performance so this is a welcome bit of research extending it"). Additionally, the autoregressive property of DocNADE allows for an easy integration of LSTM-LM. In general, our approach of introducing language concepts can be further used in other flavors of topic models.
> As suggested, we *additionally* show comparison of our proposed methods (i.e., ctx-DocNADE and ctx-DocNADEe) with NTM [1], ProdLDA [2] and SCHOLAR [3].
>
> Comparison with NTM [1]: In terms of classification accuracy on 20NS dataset, we show that both ctx-DocNADE (0.744  vs 0.72) and ctx-DocNADEe (0.751 vs 0.72) outperform NTM (results taken from [1], Figure 2). In terms of topic coherence (NPMI) on the BNC dataset (new experiments), DocNADE, ctx-DocNADE and ctx-DocNADEe outperform NTM over 50, 100 and 150 topics (see Table 7 in revised version).
>
> Comparison with ProdLDA [2]: A *direct* comparison with our work is *not fair* due to differences in its motivation (focus on improving inference in LDA) and experimental setup, for instance the vocabulary size. In order to fairly compare the DocNADE baseline  and ProdLDA in the same experimental setup as ProdLDA, we rerun DocNADE on the 20NS dataset for 200 topics and oberved that DocNADE outperformed ProdLDA, i.e., a PPL score of 830 vs 1168, which is *better* than ProdLDA [2] (result taken from Table 3 of [2]).
>
> Comparison with SCHOLAR [3]: In terms of classification accuracy, the proposed models ctx-DocNADE (0.744  vs 0.71) and ctx-DocNADEe (0.751 vs 0.71) outperform SCHOLAR ([3], Table 2) on the 20NS dataset. Importantly, SCHOLAR focuses on incorporating meta-data (author, source, date, etc.) into topic models, different to our motivation of introducing language concepts using the document text only. Note that in the submission, we had reported a corresponding F1 score of .745 by ctx-DocNADEe in Table 4.
>
> The above arguments clearly indicate that DocNADE is a *stronger* baseline than the three suggested baselines (NTM[1], ProdLDA[2] and SCHOLAR[3]). Additionally, we have quantitatively demonstrated that our proposed models ctx-DocNADE and ctx-DocNADEe outperform DocNADE.

---

> > ### Comment · AnonReviewer1 · 2018-11-21
> > **Please compare the proposed methods with PordLDA and SCHOLAR using coherence and perplexity**
> >
> > For NTM, glove-DMM, glove-LDA, and gaussian-LDA, I think your explanations are reasonable.  In their papers, they indeed do not provide both coherence and perplexity.
> >
> > However, in SCHOLAR and PordLDA, they do evaluate their methods using both coherence and perplexity in their papers. As far as I understand from the reported results of ProdLDA [2], one method could generate better coherence scores but worse perplexity by making their clusters larger (using fewer topics). Thus, to argue that your topic model is better in every aspect, you need to compare using both metrics.
> >
> > In [2,3], SCHOLAR and PordLDA are reported to have a much higher coherence and much lower perplexity than LDA. It is misleading not to report their coherences while you claim your model is better than many other models based on coherence. You could set up the fair comparison with PordLDA using perplexity, so I suppose it should not be hard to also have a fair comparison with them in terms of coherence.
> >
> > It is fine if you find the proposed methods are worse than them in terms of coherences but better in terms of perplexity as long as you could properly explain why that happened. If there is no space, please at least put the results into the supplementary materials to provide a full picture to readers.
> >
> > By the way, why are the values of perplexity in Table 5 very different from the values in Table 13?

---

> > > ### Author Response · Authors · 2018-11-23
> > > **Revised version updated with suggested methods: ProdLDA and SCHOLAR**
> > >
> > > Thanks for your comments again. The additional experiments as suggested would help in strengthening our paper.
> > >
> > > We have now revised our paper with the suggested topic models (ProdLDA and SCHOLAR) and compared them for three evaluation types: perplexity, topic coherence and information retrieval task.
> > >
> > > In our previous response, we have discussed the difference in motivation of our proposed models and the suggested topic models, and now included them in the revised version based on your feedback.
> > >
> > > Importantly in the revised version, we have included more related baselines, such as Topic-RNN, TDLM and TCNLM, as suggested by other reviewers.
> > >
> > > Please see the revised version. The updates are:
> > > 1. Figure 3f: Added IR curve for ProdLDA
> > > 2. section 3: additional evaluation including ProdLDA and SCHOLAR
> > > 3. appendices
> > >
> > > About "the values of perplexity in Table 5 very different from the values in Table 13":
> > > -----------------------------------------------------------
> > > Table 5 for PPL scores on test set
> > > Table 13 for PPL scores on validation set

---

> > > > ### Comment · AnonReviewer1 · 2018-11-23
> > > > **Thanks for your revision**
> > > >
> > > > Glad to see that this paper provides a very comprehensive comparison and a full picture to the readers. I have changed my position to acceptance as I promised originally.
> > > >
> > > > Finally, I recommend the authors to add some explanations or hypotheses which explain why some neural topic models such as ProdLDA and SCHOLAR could get a much better coherence score but a much worse perplexity score.

---

> > > > > ### Author Response · Authors · 2018-11-23
> > > > > **Thanks for raising overall score and positive comment: "very comprehensive comparison"**
> > > > >
> > > > > We appreciate that you acknowledged our additional comparisons in the revised version.
> > > > >
> > > > > We are glad to see the improved overall score and an 'accept'. Thank you for revising your scores.

---

### Public Comment · (anonymous) · 2018-10-18
**Small questions about your experimental setup and results**

Overall, the paper looks strong technically. But the experiments (on IR and Classification) do not show its strong promise:
1, In Table 3, it looks like the very simple baseline "glove" can get very high F1, and slightly lower IR. This indicates me that in practical, sum up of glove/word2vec embeddings is good enough;

2, For the "Categoriztion" experience, you used logistic regression over doc representation learned by your methods and those baselines;  so.... it is supervised system now; why not compare with the popular CNN/LSTM+LR classifiers?   If you really want to focus on unsupervised system, i guess the "Categorization" can be implemented by matching the doc and label representations, then choose the closest label; No training needed;

---

> ### Author Response · Authors · 2018-10-22
> **Small questions about your experimental setup and results**
>
> Thanks for your comments/suggestions.
>
> Please find our response below:
>
> 1.	Experimental results on IR and F1:
>
> In this work, we have introduced contextual information and word embeddings into neural autoregressive topic models (DocNADE) with ease and shown that our contributions outperform all the baseline topic models, in terms of generalization (perplexity), topic coherence, text retrieval and classification.
>
> Table 3 illustrates the scores for information retrieval (IR) and classification (F1) tasks on *short-text* datasets, quantifying the quality of representations learned by the baseline topic models (row #4-#9). The scores in rows #1-#3 are additional baselines that we used in this work to quantify text representations generated by glove and doc2vec methods.
>
> About IR, we have clearly shown a strong promise (Table 3 and 4), compared to the 'glove' baseline:
> - On an average over 8 *short-text* datasets, we (i.e., ctx-DocNADEe) show a gain of 13.7% (.630 vs .554) in precision at retrieval fraction 0.02.
> - On an average over 6 *long-text* datasets, we (i.e., ctx-DocNADEe) show a gain of 20.0% (.601 vs .501) in precision at retrieval fraction 0.02.
>
> Additionally, to quantify the quality of word representations learned, we also compare the text classification scores of our proposed method (i.e., ctx-DocNADEe) and glove embeddings. While only for the short-text, we observe that the average F1 scores over 8 short-text datasets due to glove baseline is competitive, ctx-DocNADEe outperforms glove (F1: 0.618 vs 0.575) on average over 6 long-text datasets. The behavior is obvious because "the application of topic models is challenging to short-text datasets due to a small number of word occurrences (i.e., lack of context)" and therefore, the difficulties in learning representations. However, one can exploit the glove vectors to generate more coherent topics. It is further demonstrated in the related works of topic modeling, such as DocNADE, Gauss-LDA, glove-DMM and glove-LDA (Table 3, row #4-#9). We treat them as the baselines and outperform by a noticeable margin in all evaluations.
>
> With a focus on improving topic models for unsupervised tasks, such as generalization (perplexity), text retrieval and topic extraction (i.e., coherence), our proposed models (i.e., ctx-DocNADE and ctx-DocNADEe) outperform all the baselines for both the short-text and long-text datasets, where the topic models extract topics, encoding the thematic structures in documents that explains them. On other hand, the glove vectors encode semantic similarity in word representations.
>
> 2.	In this work, we focus on unsupervised systems in context of topic modeling.  Following the experimental setup of RSM (Salakhutdinov and Hinton, 2009) and DocNADE (Larochelle and Lauly, 2012; Lauly et al., 2017), we perform the task of IR and report precision at different retrieval fractions. The IR task is similar to the one suggested.

---

### Public Comment · ~Adji_Bousso_Dieng1 · 2018-10-18
**How about citing and comparing to recent work?**

Hi there,

You have not mentioned any of the many recent work on marrying topic models and autoregressive neural models. There are just too many for you to not cite none of them. Please correct this in your revision; respecting related work is important. I added the main references below for your information.

https://ieeexplore.ieee.org/document/6424228
https://arxiv.org/abs/1611.01702
https://arxiv.org/abs/1712.09783
https://arxiv.org/abs/1704.08012

---

> ### Author Response · Authors · 2018-10-26
> **citing and comparing to recent work**
>
> We appreciate your comments about missing citations of related neural topic models (TMs) and we will make sure to include them in a revised version. While in our initial submission we considered a comparison of our model to these studies as out of scope, we have since been running additional experiments to quantitatively compare TM performance of our proposed models to the studies you mention. We will include these new comparisons in the revised version.
>
> In the following, we briefly summarize our contributions again and contrast them to the related work you mention. We then present the results of our new experiments and show that DocNADE-based models have a competitive or superior performance in terms of topic coherence, compared to Topic-RNN, TCNLM and TDLM.
>
> The related studies you mention focus on improving language models (LMs) using TMs: in contrast, the focus of our work is on improving TM for textual representations (short-text or long-text documents) by incorporating language concepts (e.g., word ordering, syntax, semantics, etc.) via neural LMs. In addition, we also address the challenges of topic learning that arise from limited context in collections of "short-text" and/or  "few" documents. Therefore, as part of our second contribution, we incorporate external knowledge, i.e., word embeddings into neural autoregressive TM to better model such document collections. In contrast to this sparse-data setting, the related approaches you mention were designed for collections of long-text as well as a sufficient number of documents.
>
> We have already compared our contributions to several baselines (e.g., DocNADE, glove, glove-DMM, glove-LDA, gaussian-LDA, doc2vec, etc.) that either
> (1) do not model contextualized word information, or
> (2) do not incorporate word embeddings, or
> (3) ignore both in TMs.
> We outperform them in terms of perplexity, topic coherence, text retrieval and classification, evaluated on 14 datasets.
>
> We now further compare our models to other approaches combining TMs with autoregressive neural models. To this end, we follow the approach presented in "Topic Compositional Neural Language Model" (Wang et al., 2018) and quantitatively compare TM performance in terms of topic coherence (NPMI) on the BNC dataset.
>
> Model				Topic Coherence (BNC dataset)
> 				         #Topics   =50    =100	 =150
> ------------------------------------------------------------------------
> LDA# 						.106    .119     .119
> NTM# 						.081    .070     .072
> TDLM(s)# 					.102    .106	.100
> TDLM(l)#    				        .095	    .101	.104
> Topic-RNN(s)#                              .102    .108     .102
> Topic-RNN(l)#                               .100    .105     .097
> TCNLM(s)#                                    .114    .111     .107
> TCNLM(l)#                                     .101    .104     .102
>
>                           (sliding window=20)
> DocNADE                                       .097    .095     .097
> ctx-DocNADE*(lambda=0.2)      .102    .103     .102
> ctx-DocNADE*(lambda=0.8)      .106    .105     .104
> ctx-DocNADEe*(lambda=0.2)    .098    .101       -
> ctx-DocNADEe*(lambda=0.8)    .105    .104       -
>
>                          (sliding window=110)
> DocNADE                                       .133    .131     .132
> ctx-DocNADE*(lambda=0.2)      .134     .141     .138
> ctx-DocNADE*(lambda=0.8)      .139    .142     .140
> ctx-DocNADEe*(lambda=0.2)    .133    .139       -
> ctx-DocNADEe*(lambda=0.8)    .135    .141       -
>
>
> Here, the asterisk (*) indicates our proposed models and (#) taken from Wang et al.; lambda is the mixture weight of the LM component in the topic modeling process and sliding window is one of the hyper-parameters for computing topic coherence (Wang et al. (2018) and "Exploring the Space of Topic Coherence Measures" (Röder et al., 2015). A sliding window of 20 is used in Wang et al.; in addition we also present results for a window of size 110.
>
> Our results (table above) suggest that our contribution (i.e., ctx-DocNADE) of introducing language concepts into bag-of-word topic model (i.e., DocNADE) improves topic coherence. Better performance for high values of lambda illustrate the relevance of the LM component for topic coherence (DocNADE corresponds to lambda=0). Similarly, the inclusion of word embeddings (i.e., ctx-DocNADEe) results in more coherent topics than the baseline DocNADE.
>
> Importantly, while ctx-DocNADEe was motivated by sparse data settings, the BNC dataset is neither a collection of short-text nor a corpus of few documents. Consequently, ctx-DocNADEe does not show improvements in topic coherence over ctx-DocNADE.
>
> Beyond topic coherence, we also evaluate TMs for text retrieval as well as classification tasks on 14 datasets. However, since the BNC dataset is unlabeled, we are here restricted to comparing model performance in terms of topic coherence only.
>
> We will include the additional results (above) in the revised version.

---

> > ### Author Response · Authors · 2018-10-26
> > **citing and comparing to recent work**
> >
> > In continuation to our response above, we further qualitatively show the top 5 words of six learnt topics (topic name summarised by Wang et al., 2018):
> >
> > ---------------------------------------------------------------------------------------------------------------------------
> > 	Topic			Model        	    Topic-words (ranked by probabilities)
> > ---------------------------------------------------------------------------------------------------------------------------
> > 				TCNLM#			  pollution, emissions, nuclear, waste, environmental
> > environment		ctx-DocNADE*       ozone, pollution, emissions, warming, waste
> >                                 ctx-DocNADEe*     pollution, emissions, dioxide, warming, environmental
> > ---------------------------------------------------------------------------------------------------------------------------
> >                                TCNLM#                   elections, economic, minister, political, democratic
> > politics                  ctx-DocNADE*        elections, democracy, votes, democratic, communist
> >                                ctx-DocNADEe*      democrat, candidate, voters, democrats, poll
> > ---------------------------------------------------------------------------------------------------------------------------
> >                                TCNLM#                   album, band, guitar, music, film
> > art                         ctx-DocNADE*         guitar, album, band, bass, tone
> >                               ctx-DocNADEe*       guitar, album, pop, guitars, song
> > ---------------------------------------------------------------------------------------------------------------------------
> >                               TCNLM#                    bedrooms, hotel, garden, situated, rooms
> > facilities               ctx-DocNADE*          bedrooms, queen, hotel, situated, furnished
> >                               ctx-DocNADEe*       hotel, bedrooms, golf, resorts, relax
> > ---------------------------------------------------------------------------------------------------------------------------
> >                              TCNLM#                    corp, turnover, unix, net, profits
> > business             ctx-DocNADE*          shares, dividend, shareholders, stock, profits
> >                              ctx-DocNADEe*        profits, growing, net, earnings, turnover
> > ---------------------------------------------------------------------------------------------------------------------------
> >                              TCNLM#                     eye, looked, hair, lips, stared
> > expression         ctx-DocNADE*           nodded, shook, looked, smiled, stared
> >                             ctx-DocNADEe*         charming, smiled, nodded, dressed, eyes
> > ---------------------------------------------------------------------------------------------------------------------------
> >
> > We will include the topics (above) in the revised version.

---

### Public Comment · (anonymous) · 2018-10-21
**Few Comments/Questions**

Very interesting work (improving topic modelling using RNN), the paper looks quite strong technically, I believe it will be helpful for the community. I especially like the idea of using compositional distributional priors which seems effective for short text passages. Also, the results are very nice,  especially for IR. I have some comments/questions:
i. Will you release the source code?
ii. reference is duplicated for 'Deep temporal-recurrent-replicated-softmax for topical trends over time.'
ii. page 2, (last line) However, in the distributed embedding space, the word pairs are semantically related as shown in Figure 1 (left) -> shouldn't it be Figure 1 (right)?

---

> ### Author Response · Authors · 2018-10-22
> **Few Comments/Questions**
>
> Thanks for your positive comments.
>
> We will incorporate your suggestions in our revised version.
>
> Yes, we will release the code and pre-processed datasets.

---

### Public Comment · (anonymous) · 2018-10-24
**Some clarifications**

Interesting work! I do like the idea of word-order cues into topic models. A few clarifications:
(a) You mention DocNade does not model context since it computes Eqn (1) using word ID's < i (which can be arbitrary)? What happens if you still use DocNade exactly as it is, but just modify that computation in Eqn 1 to use only the token ID's corresponding to the context. This approach would be a reasonable baseline without using an LSTM. Perhaps you can justify the superiority of your model over this?

(b) Would your method also explicitly represent a document as a probability distribution over topics or is it assumed that the latent representation does not necessarily imply a probability distribution?

---

> ### Author Response · Authors · 2018-10-26
> **Some clarifications**
>
> Thanks for your interest and comments about our work.
>
> Please find our answers below:
>
> (a) As mentioned in Larochelle & Lauly (2012), due to the bag-of-word nature of DocNADE, a document v  takes the form of a set of word counts vectors in which the original word order is lost; this, in turn, is required by DocNADE to compute the sequence of conditionals p(v_i | v<i). Consequently, a random permutation of its words is generated in each iteration an, as a result, a distribution over all possible permutations that could have generated the original document v is obtained  (see also section 4.1 in Larochelle & Lauly, 2012).
> While in principle it is an interesting approach to use only those token IDs corresponding to the context of a given target word v_i, it is not possible to obtain the token IDs in the document, given the word IDs from its bag-of-word representation. For instance, given a word ID that occurs n times, it is not feasible to trace back the token ID(s) in the document.
>
> We would also like to add that the neural language model not only introduces word ordering into topic models, but also language concepts such as the syntax and semantics encoded in its internal states ("Deep contextualized word representations", Peters et al. (2018)).
>
> (b) Yes. Following the baseline topic model (i.e., DocNADE), our modeling approaches explicitly do represent a document as a probability distribution over topics. This can be seen in equations (1) and (2) in the paper, where the autoregressive conditional p(v_i = w | v_<i) computes H (i.e., number of topics) distributions over the vocabulary for the input document v, using the H dimensional hidden topic vector.

---

### Public Comment · (anonymous) · 2018-10-27
**Interesting work**

I liked the concept. A few quick questions.

1. How efficient is the vector generator in terms of ease of use and implementation as compared to existing state-of-the-art techniques?

2. Did you release the source code and pre-trained model for community evaluation?

I see you tested on an array of datasets.

---

> ### Author Response · Authors · 2018-10-29
> **interesting work**
>
> Thanks for your interest in our work, positive comments and questions.
>
> Please find our response below:
>
> 1.  We briefly discussed the model complexity in section 2.2.
>
> 2. Yes, we will release the source code and also the pre-trained models.

---

### Author Response · Authors · 2018-11-20
**Revised version with suggested related works and experimental results**


We thank all the official and non-official reviewers for your insightful comments.

We are glad to hear positive comments, especially from the official reviewers, such as "solid and significantly better results", "extensive experiments", "four different kinds of evaluation is good", "commendable job in thoroughly evaluating", etc.

Based on your feedbacks, we have revised our paper with additional experimental results including in total **6 additional baselines** and some minor comments about presentation.  We have also addressed specific questions from each of the official reviewers.

Remark about suggested baselines:
---------------------------------
We have run additional experiments to cover the suggested baselines, such as NTM, ProdLDA, SCHOLAR, TDLM, Topic-RNN and TCNLM. We have shown that our proposed models outperform or are competitive to the baselines. Importantly, the related studies (e.g., TDLM, Topic-RNN and TCNLM) in marrying topic and language models is an active area of research and primarily focus on improving language modeling tasks using topic models: in contrast, the focus of our work is on improving topic models for textual representations (short-text or long-text documents) by incorporating language concepts (e.g., word ordering, syntax, semantics, etc.) via neural language models. In addition, we have also addressed the challenges of topic learning that arise from limited context in collections of "short-text" and/or  "few" documents. Therefore, as part of our second contribution, we incorporate external knowledge, i.e., word embeddings into neural autoregressive topic model to better model such document collections. In contrast to this sparse-data setting, the related approaches were designed for collections of long-text as well as a sufficient number of documents.

In spite of the differences with the related studies (as suggested), we have now run additional experiments using these baselines: IR task using TDLM on 8 short-text datasets and topic coherence (NPMI) using an additional dataset to compare our models (i.e., ctx-DocNADE and ctx-DocNADEe) with NTM, TDLM, Topic-RNN and TCNLM. In the revised version, we have further shown that our proposed models also outperform TDLM by a noticeable margin in terms of text retrieval and categorization for all the 8 short-text datasets.


To summarize, the updates in revised version:
1. [New] Table 7 of additional experiments, as suggested.
2. [New] Figure 4
3. [Update] Table 3
4. [Update] Figure 3 (a,b,c,d and e)
5. [Update] section 3.2
6. [Update] section 3.3 (for additional experiments)
7. [Update] section 3.4 (additional paragraph for new experiments)

Remark: Diff in the two versions looks significant (*although it is not*) due to re-positioning of figures and tables. The table 7 is added after Table 6 and figure 3 (which is before Table 6 in the submission version) is now moved after Table 7  in the revised version.

*** Please see the revised version ****

---

### Author Response · Authors · 2018-11-23
**Revised version updated with additional comparisons with ProdLDA and SCHOLAR**

Please see the revised version.

The updates are:
1. Figure 3f: Added IR curve for ProdLDA
2. section 3: additional evaluation including ProdLDA and SCHOLAR
3. appendices

---

### Author Response · Authors · 2019-02-23
**Camera Ready Uploaded**

Camera Ready Uploaded !!

---

### Meta-Review · Area_Chair1 · 2018-12-14
**While not groundbreaking, the proposed model is new and the empirical results are strong.**

**Confidence:** 3
**Recommendation:** Accept (Poster)

**Metareview:**

This paper presents an extension of an existing topic model, DocNADE. Compared to DocNADE and other existing bag-of-word topic models, the primary contribution of this work is to integrate neural language models into the topic model in order to address two limitations of the bag-of-word topic models: expressiveness and interpretability. In addtion, the paper presents an approach to integrate external knowledge into the neural topic models to address the empirical challenges of the application scenarios where there might be only a small training corpus or limited context available.

Pros:
The paper presents strong and extensive empirical results. The authors went above and beyond to strengthen their paper during the rebuttal and address all the reviewers' questions and suggestions (e.g., the submitted version had 7 baselines, and the revised version has 6 additional baselines per reviewers' requests).

Cons:
The paper builds on an earlier paper that introduced the DocNADE model. Thus, the modeling contribution is relatively marginal. On the other hand, the extended model, albeit based on a relatively simple idea, is still new and demonstrates strong empirical results.

Verdict:
Probably accept. While not groundbreaking, the proposed model is new and the empirical results are strong.